# FORGETTING TRANSFORMER: SOFTMAX ATTENTION WITH A FORGET GATE

**Zhixuan Lin**[*]
Mila & Université de Montréal
zxlin.cs@gmail.com

**Evgenii Nikishin**
Mila & Université de Montréal
evgenii.nikishin@mila.quebec

**Xu Owen He**[†]
MakerMaker AI
owen.hexu@gmail.com

**Aaron Courville**
Mila & Université de Montréal
courvila@mila.quebec

## ABSTRACT

An essential component of modern recurrent sequence models is the *forget gate*. While Transformers do not have an explicit recurrent form, we show that a forget gate can be naturally incorporated into Transformers by down-weighting the unnormalized attention scores in a data-dependent way. We name this attention mechanism *Forgetting Attention* and the resulting model the *Forgetting Transformer (FoX)*. We show that FoX outperforms the Transformer on long-context language modeling, length extrapolation, and short-context downstream tasks, while performing on par with the Transformer on long-context downstream tasks. Moreover, it is compatible with the FlashAttention algorithm and does not require any positional embeddings. Several analyses, including the needle-in-the-haystack test, show that FoX also retains the Transformer's superior long-context capabilities over recurrent sequence models such as Mamba-2, HGRN2, and DeltaNet. We also introduce a "Pro" block design that incorporates some common architectural components in recurrent sequence models and find it significantly improves the performance of both FoX and the Transformer. Our code is available at `https://github.com/zhixuan-lin/forgetting-transformer`.

## 1 INTRODUCTION

Despite the growing interest in reviving recurrent sequence models (Gu et al., 2021; Peng et al., 2021; Yang et al., 2023; Gu & Dao, 2023; Sun et al., 2023; De et al., 2024; Qin et al., 2024b; Dao & Gu, 2024; Peng et al., 2024; Beck et al., 2024; Zhang et al., 2024), these models still underperform the Transformer (Vaswani et al., 2017) in terms of *long-context capabilities* (Hsieh et al., 2024; Waleffe et al., 2024; Shen et al., 2024; Qin et al., 2024a), likely due to their relatively small fixed-sized hidden states (Jelassi et al., 2024). While the Transformer excels in handling long-context information, it lacks an explicit mechanism for forgetting past information in a *data-dependent* way. Such a mechanism – often implemented as some form of the *forget gate* (Gers et al., 2000) – is ubiquitous in recurrent sequence models and has proven critical in their success in short-context tasks (Greff et al., 2016; Van Der Westhuizen & Lasenby, 2018; Peng et al., 2021; Yang et al., 2023; Gu & Dao, 2023). A natural question to ask is then: can we have a forget gate in Transformers?

To address this question, we leverage an important fact: many recurrent sequence models with a forget gate can be written in a parallel linear attention form (Katharopoulos et al., 2020) analogous to softmax attention (Yang et al., 2023; Dao & Gu, 2024). In this parallel form, the forget gate mechanism translates into down-weighing the unnormalized attention scores in a data-dependent way. Our key insight is that this *exact* mechanism is also applicable to softmax attention. We name this attention mechanism *Forgetting Attention* and the resulting model the **Forgetting Transformer (FoX)**.

---

[*]Correspondence to Zhixuan Lin.
[†]Work done while at Google DeepMind.

We show that FoX outperforms the Transformer on long-context language modeling, length extrapolation, and short-context downstream tasks, while performing on par with the Transformer on long-context downstream tasks. Notably, it does not require any positional embeddings. It also retains Transformers' long-context retrieval abilities and achieves near-perfect accuracy in the needle-in-the-haystack test (Kamradt, 2023) within the training context length. In contrast, all the tested recurrent sequence models fail. We also introduce a "Pro" block design that integrates several architectural components commonly used in recurrent sequence models, which significantly improves the performance of FoX and the baseline Transformer. Finally, we show that FoX can be implemented in a hardware-aware way with a simple modification to the FlashAttention (Dao, 2023) algorithm.

## 2 BACKGROUND: LINEAR ATTENTION WITH A FORGET GATE

This section introduces the notation used in this work and gives a brief background on linear attention. We also introduce a gated variant of linear attention and discuss its parallel form, which naturally leads to FoX. Throughout this work, we only consider causal sequence modeling.

### 2.1 LINEAR ATTENTION

Standard causal softmax attention takes a sequence of input vectors $(\boldsymbol{x}_i)_{i=1}^L$ and produces a sequence of output vectors $(\boldsymbol{o}_i)_{i=1}^L$, where $\boldsymbol{x}_i, \boldsymbol{o}_i \in \mathbb{R}^d, i \in \{1, \dots, L\}$. Each $\boldsymbol{o}_i$ is computed as follows:

$$\boldsymbol{q}_i, \boldsymbol{k}_i, \boldsymbol{v}_i = \boldsymbol{W}_q \boldsymbol{x}_i, \boldsymbol{W}_k \boldsymbol{x}_i, \boldsymbol{W}_v \boldsymbol{x}_i \in \mathbb{R}^d, \tag{1}$$

$$\boldsymbol{o}_i = \frac{\sum_{j=1}^i k_{\exp}(\boldsymbol{q}_i, \boldsymbol{k}_j) \boldsymbol{v}_j}{\sum_{j=1}^i k_{\exp}(\boldsymbol{q}_i, \boldsymbol{k}_j)} = \frac{\sum_{j=1}^i \exp(\boldsymbol{q}_i^\top \boldsymbol{k}_j) \boldsymbol{v}_j}{\sum_{j=1}^i \exp(\boldsymbol{q}_i^\top \boldsymbol{k}_j)}, \tag{2}$$

where $\boldsymbol{W}_q, \boldsymbol{W}_k, \boldsymbol{W}_v \in \mathbb{R}^{d \times d}$ are projection matrices and $k_{\exp}(\boldsymbol{q}, \boldsymbol{k}) = \exp(\boldsymbol{q}^\top \boldsymbol{k})$ is the exponential dot product kernel.[1]

Linear attention (Katharopoulos et al., 2020) replaces the exponential dot product kernel $k_{\exp}(\boldsymbol{q}, \boldsymbol{k}) = \exp(\boldsymbol{q}^\top \boldsymbol{k})$ with a kernel $k_\phi(\boldsymbol{q}, \boldsymbol{k})$ with some feature representation $\phi : \mathbb{R}^d \to (\mathbb{R}^+)^{d'}$:

$$\boldsymbol{o}_i = \frac{\sum_{j=1}^i k_\phi(\boldsymbol{q}_i, \boldsymbol{k}_j) \boldsymbol{v}_j}{\sum_{j=1}^i k_\phi(\boldsymbol{q}_i, \boldsymbol{k}_j)} = \frac{\sum_{j=1}^i (\phi(\boldsymbol{q}_i)^\top \phi(\boldsymbol{k}_j)) \boldsymbol{v}_j}{\sum_{j=1}^i \phi(\boldsymbol{q}_i)^\top \phi(\boldsymbol{k}_j)} \tag{3}$$

Following Yang et al. (2023), we call this the *parallel form* of linear attention as it can be computed with matrix multiplications. Alternatively, linear attention can be computed in a *recurrent form*:

$$\boldsymbol{S}_t = \boldsymbol{S}_{t-1} + \boldsymbol{v}_t \phi(\boldsymbol{k}_t)^\top \tag{4}$$

$$\boldsymbol{z}_t = \boldsymbol{z}_{t-1} + \phi(\boldsymbol{k}_t) \tag{5}$$

$$\boldsymbol{o}_t = \frac{\boldsymbol{S}_t \phi(\boldsymbol{q}_t)}{\boldsymbol{z}_t^\top \phi(\boldsymbol{q}_t)}, \tag{6}$$

where $\boldsymbol{S}_t \in \mathbb{R}^{d \times d'}, \boldsymbol{z}_t \in \mathbb{R}^{d'}, t \in \{0, \dots, L\}$ are computed recurrently, with $\boldsymbol{S}_0 = \boldsymbol{0}$ and $\boldsymbol{z}_t = \boldsymbol{0}$.

### 2.2 LINEAR ATTENTION WITH A FORGET GATE

The recurrent form of linear attention makes it natural to introduce a forget gate. Specifically, we can compute a scalar forget gate $f_t = \sigma(\boldsymbol{w}_f^\top \boldsymbol{x}_t + b_f) \in \mathbb{R}$ at each timestep, where $\sigma$ is the sigmoid function and $\boldsymbol{w}_f \in \mathbb{R}^d, b_f \in \mathbb{R}$ are learnable parameters. The recurrent computation is then:

$$\boldsymbol{S}_t = f_t \boldsymbol{S}_{t-1} + \boldsymbol{v}_t \phi(\boldsymbol{k}_t)^\top \tag{7}$$

$$\boldsymbol{z}_t = f_t \boldsymbol{z}_{t-1} + \phi(\boldsymbol{k}_t) \tag{8}$$

$$\boldsymbol{o}_t = \frac{\boldsymbol{S}_t \phi(\boldsymbol{q}_t)}{\boldsymbol{z}_t^\top \phi(\boldsymbol{q}_t)}. \tag{9}$$

---

[1] Note we omit the $\frac{1}{\sqrt{d}}$ scaling factor to reduce visual clutter. In practice we always scale $\boldsymbol{q}_i^\top \boldsymbol{k}_j$ by $\frac{1}{\sqrt{d}}$.

Note that this gated variant of linear attention differs from most models in the literature. In particular, most gated variants of linear attention models, such as GLA (Yang et al., 2023) and Mamba-2 (Dao & Gu, 2024), do not have the normalization term (i.e., there is no $z_t$, and the output is just $o_t = S_t \phi(q_t)$). We keep the normalization term to maintain similarity with softmax attention.

Crucially, similar to the normalization-free version derived in GLA and Mamba-2, we can show that this gated variant of linear attention also has a parallel form:

$$o_i = \frac{\sum_{j=1}^{i} F_{ij} \phi(q_i)^\top \phi(k_j) v_j}{\sum_{j=1}^{i} F_{ij} \phi(q_i)^\top \phi(k_j)} = \frac{\sum_{j=1}^{i} F_{ij} k_\phi(q_i, k_j) v_j}{\sum_{j=1}^{i} F_{ij} k_\phi(q_i, k_j)}, \tag{10}$$

where $F_{ij} = \prod_{l=j+1}^{i} f_l$, with the convention that $F_{ij} = 1$ if $i = j$. Our key observation is that Equation 10 and the softmax attention in Equation 2 are very similar in form. In fact, if we change the kernel $k_\phi$ in Equation 10 back to the exponential dot product kernel $k_{\exp}$, we obtain *softmax attention with a forget gate*. We introduce this formally in the next section.

## 3 FORGETTING TRANSFORMER

Our proposed model, the ***Forgetting Transformer (FoX)***, features a modified softmax attention mechanism with a forget gate. We name this attention mechanism *Forgetting Attention*. Similar to the gated variant of linear attention introduced in the previous section, we first compute a scalar forget gate $f_t = \sigma(w_f^\top x_t + b_f) \in \mathbb{R}$ for each timestep $t$. The output of the attention is then

$$o_i = \frac{\sum_{j=1}^{i} F_{ij} \exp(q_i^\top k_j) v_j}{\sum_{j=1}^{i} F_{ij} \exp(q_i^\top k_j)} = \frac{\sum_{j=1}^{i} \exp(q_i^\top k_j + D_{ij}) v_j}{\sum_{j=1}^{i} \exp(q_i^\top k_j + D_{ij})}, \tag{11}$$

where $F_{ij} = \prod_{l=j+1}^{i} f_l$ and $D_{ij} = \log F_{ij} = \sum_{l=j+1}^{i} \log f_l$. This can be written in matrix form:

$$D = \log F \in \mathbb{R}^{L \times L}, \tag{12}$$

$$O = \mathrm{softmax}(QK^\top + D)V \in \mathbb{R}^{L \times d}, \tag{13}$$

where $F \in \mathbb{R}^{L \times L}$ is a lower triangular matrix whose non-zero entries are $F_{ij}$, i.e., $F_{ij} = F_{ij}$ if $i \geq j$ and 0 otherwise. We adopt the convention that $\log 0 = -\infty$. $Q, K, V, O \in \mathbb{R}^{L \times d}$ are matrices containing $q_i, k_i, v_i, o_i, i \in \{1, \dots, L\}$ as the rows. The $\mathrm{softmax}$ operation is applied row-wise. For multi-head attention with $H$ heads, we maintain $H$ instances of forget gate parameters $\{w_f^{(h)}\}_{h=1}^{H}$ and $\{b_f^{(h)}\}_{h=1}^{H}$ and compute the forget gate values $\{f_t^{(h)}\}_{h=1}^{H}$ separately for each head.

**Hardware-aware implementation**  The logit bias form on the rightmost side of Equation 11 can be computed with a simple modification to the FlashAttention (Dao, 2023) algorithm. Here we briefly describe the forward pass. The backward pass follows a similar idea. First, we compute the cumulative sum $c_i = \sum_{l=1}^{i} \log f_l$ for $i \in \{1, \dots, L\}$ and store it in the high-bandwidth memory (HBM) of the GPU. This allows us to compute $D_{ij} = c_i - c_j$ easily later. Whenever we compute the attention logit $q_i^\top k_j$ in the GPU's fast shared memory (SRAM) (as in FlashAttention), we also load $c_i$ and $c_j$ to SRAM, compute $D_{ij}$, and add it to the attention logit. The rest of the forward pass remains the same as FlashAttention. This algorithm avoids instantiating the $L \times L$ $D$ matrix in the HBM. We provide a detailed algorithm description in Appendix E. Moreover, since the forget gates are scalars instead of vectors, the additional computation and parameters introduced are negligible.

**Connection to ALiBi**  Besides its natural connection to gated linear attention, Forgetting Attention can also be seen as a data-dependent and learnable version of ALiBi (Press et al., 2021). ALiBi applies a data-*independent* bias $b_{ij} = -(i - j)m_h$ to the attention logits, where $m_h$ is a fixed slope specific to each head $h$. It is easy to show that ALiBi is equivalent to using a fixed, head-specific, and data-independent forget gate $f_t^{(h)} = \exp(-m_h)$. In Section 4.5, we verify the superiority of data-dependent forget gates over ALiBi.

**Positional embeddings**  Though we find that using Rotary Position Embeddings (RoPE) (Su et al., 2024) sometimes slightly improves the performance of FoX, it is not necessary as it is for the standard Transformer (see ablations in Section 4.5). For simplicity, *we do not use RoPE or any other positional embeddings for FoX by default.*

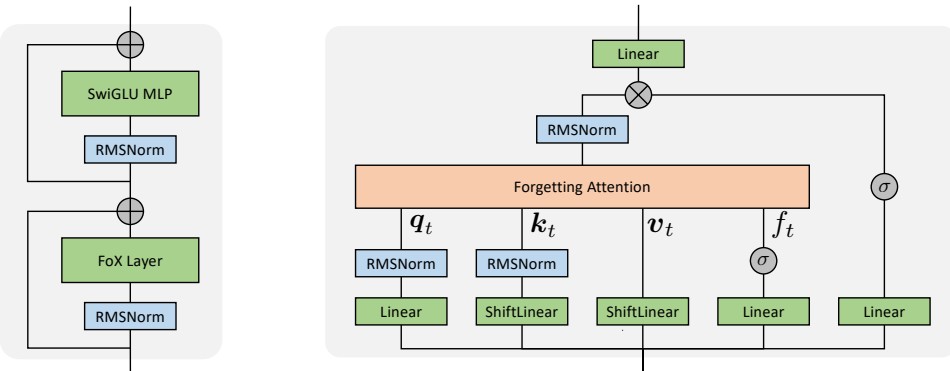

Figure 1: Default architecture of FoX. (**left**) A single FoX block. (**right**) A single FoX (Pro) layer. All RMSNorms on the right are applied independently to each head. $\sigma$ is the sigmoid function. $\otimes$ is element-wise multiplication. `ShiftLinear` implements the computation in Equation 14.

**Architecture design**   We test FoX with two different architectures. First, we replace RoPE in the LLaMA architecture (Touvron et al., 2023) with forget gates and refer to this model as *FoX (LLaMA)*. Second, we test an improved "Pro" architecture with output gates[2] and output normalization (also used in GLA and Mamba-2). We also use QK-norm (Dehghani et al., 2023) and apply a simplified variant of data-dependent token shift (Peng et al., 2024) to the keys and values (KV-shift). Concretely, the keys $(\boldsymbol{k}_i)_{i=1}^{L}$ are computed as follows with additional parameters $\boldsymbol{w}_k \in \mathbb{R}^d$:

$$\tilde{\boldsymbol{k}}_t = \boldsymbol{W}_k \boldsymbol{x}_t \in \mathbb{R}^d, \quad \alpha_t^{\text{key}} = \sigma(\boldsymbol{w}_k^\top \boldsymbol{x}_t) \in \mathbb{R}$$
$$\boldsymbol{k}_t = \text{RMSNorm}(\alpha_t^{\text{key}} \tilde{\boldsymbol{k}}_{t-1} + (1 - \alpha_t^{\text{key}})\tilde{\boldsymbol{k}}_t) \tag{14}$$

The values $(\boldsymbol{v}_i)_{i=1}^{L}$ are computed in the same way, but without RMSNorm. The overall architecture is shown in Figure 1 and detailed in Appendix A. We refer to the resulting model as *FoX (Pro)*.

## 4   EMPIRICAL STUDY

The advantages of Transformers in long-context abilities over recurrent sequence models have been verified multiple times (Hsieh et al., 2024; Waleffe et al., 2024; Shen et al., 2024; Qin et al., 2024a). However, forget gates introduce a *recency bias*. It is thus natural to ask whether FoX still maintains this advantage. Therefore, our empirical study places a special focus on long-context capabilities.

### 4.1   EXPERIMENTAL SETUP

**Dataset and baselines**   We focus on long-context language modeling and train all models on LongCrawl64 (Buckman, 2024), a long-sequence subset of RedPajama-v2 (Together Computer, 2023) pre-tokenized with the TikToken tokenizer (OpenAI, 2022) for GPT-2 (Radford et al., 2019). For baselines, we focus on two types of comparisons. First, we compare FoX with the Transformer. For the Transformer, we also test both the LLaMA and the Pro architecture (referred to as *Transformer (LLaMA)* and *Transformer (Pro)*, respectively). Similar to Xiong et al. (2023), we find it crucial to use a large RoPE angle $\theta$ for the Transformer for long-context training. Following Xiong et al. (2023) we use $\theta = 500000$. Second, to show the advantage of FoX over recurrent sequence models in long-context capabilities, we compare it with Mamba-2 (Dao & Gu, 2024), HGRN2 (Qin et al., 2024a), and DeltaNet (Yang et al., 2024). The implementation of all models is based on the Flash Linear Attention repository (Yang & Zhang, 2024).

**Training setup**   For our main experiments, we train models with 760M (non-embedding) parameters on a $45 \times 2^{30}$-token (roughly 48B tokens) subset of LongCrawl64 with a training context length

---

[2]When output gates are used, we reduce the number of parameters in the MLPs so the total number of parameters remains the same.

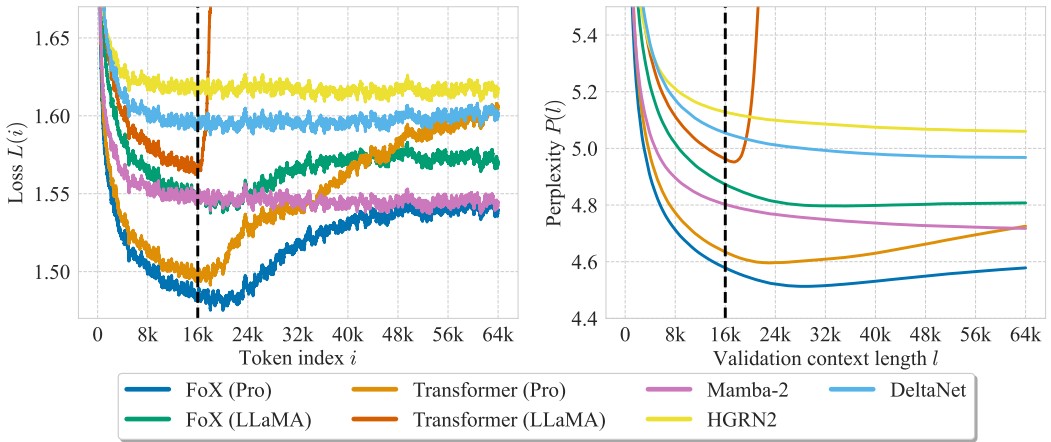

Figure 2: (**left**) Per-token loss $L(i)$ at different token position $i$. (**right**) Validation perplexity $P(l)$ over different validation context length $l$. The vertical dashed line indicates the training context length. The per-token loss is smoothed using a moving average sliding window of 101 tokens.

of 16384 tokens. For the validation set, we use a $2 \times 2^{30}$-token subset of the LongCrawl64 held-out set with sequences of 65536 tokens. We choose a longer validation context length than the training context length to test the length extrapolation abilities of the models. All models are trained with AdamW (Loshchilov, 2017) with $(\beta_1, \beta_2) = (0.9, 0.95)$. We use a linear learning rate warmup from 0 to the peak learning rate for the first $256 \times 2^{20}$ tokens and then a cosine decay to 0. Each training batch contains $0.5 \times 2^{20}$ tokens. All models use a weight decay of $0.1$ and gradient clipping of $1.0$. We search the learning rate for each model within $\{1 \times 10^i, 2 \times 10^i, 5 \times 10^i\}$ for different $i$'s until we identify a locally optimal value. We tune the head dimensions for FoX and the Transformer in $\{64, 128\}$. We find that *FoX often prefers higher learning rates and more heads/smaller head dimensions than the Transformer, and the Pro models often prefer higher learning rates than the LLaMA models*. Details of the hyperparameters and experimental setup can be found in Appendix B.

## 4.2 Long-Context Language Modeling

**Metrics** For our main metric, we use *per-token* loss on the validation set at different token positions. To be precise, let $V$ be the vocabulary size, $\boldsymbol{y}_i^{(j)} \in \{0,1\}^V$ be the one-hot vector encoding the language modeling target for the $i$-th token in the $j$-th validation sequence, and $\boldsymbol{p}_i^{(j)} \in \mathbb{R}^V$ be the corresponding output probabilities of the model, then the per-token loss $L(i)$ at token position $i$ is defined as $L(i) = \frac{1}{M} \sum_{j=1}^M -\log[(\boldsymbol{p}_i^{(j)})^\top \boldsymbol{y}_i^{(j)}]$, where $M$ is the number of validation sequences.

The per-token loss is particularly meaningful for understanding the long-context capabilities of a model. Informally, a monotonically decreasing $L(i)$ with a steep slope indicates the model is using the full context well. On the other hand, if $L(i)$ plateaus after some token position $k$, it indicates the model struggles to use tokens that are $k$ tokens away from the current token position for its prediction. This correspondence between the slope of $L(i)$ and the model's context utilization is explained in more detail in Appendix C.

Besides per-token loss, we also report perplexity over different context lengths. Concretely, perplexity $P(l)$ over a context length $l$ is defined as $P(l) = \exp(\frac{1}{l} \sum_{i=1}^l L(i))$. *We warn the readers that the slope of $P(l)$ is less meaningful*. Since $P(l)$ is the exponential of the cumulative average of $L(i)$, even if $L(i)$ plateaus after some token position $k$, $P(l)$ may still keep decreasing after $k$, giving the wrong impression that the model can make use of the part of the context that is $k$ tokens away.

**Results** In Figure 2, we show the per-token loss $L(i)$ at different token indices $i$ and perplexity $P(l)$ over different validation context lengths $l$. As shown in Figure 2, with either architecture, FoX outperforms the standard Transformer both within and beyond (i.e., length extrapolation) the

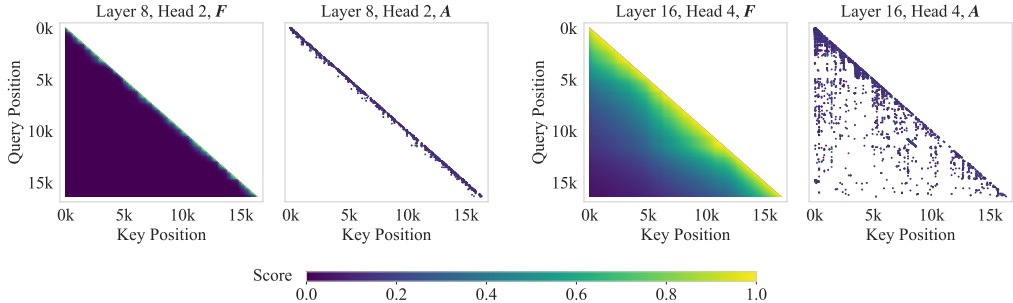

Figure 3: Visualization of the forget gate weight matrix $F$ and the attention score matrix $A$ from two heads in different layers. Since $A$ is very sparse, we only show entries with scores larger than 0.1. These results use FoX (Pro). More examples can be found in Appendix F.10.

training context length. Similar to the Transformer, it maintains a monotonically decreasing per-token loss *within the training context length*, indicating that it utilizes the entire training context for its prediction. In contrast, the per-token loss curves of all recurrent sequence models start flattening at around 5k tokens and plateau after 10k tokens. This indicates that these recurrent models struggle to use the full context effectively for their prediction. In terms of the *absolute* values of the loss and perplexity, FoX (Pro) also clearly outperforms HGRN2, DeltaNet and Mamba-2.

**Visualization of forget gate values and attention map** In Figure 3, we visualize the forget gate weight matrix $F$ and the attention scores $A = \mathrm{softmax}(QK^\top + \log F)$ from two heads in different layers. The head on the left-hand side exhibits strong decay, and most entries of $F$ are close to zero; accordingly, the attention focuses on local entries. The head on the right-hand side has much weaker decay, and the attention is distributed across the entire context. This shows that FoX can learn to retain information across long contexts when necessary.

### 4.3 NEEDLE IN THE HAYSTACK

The needle-in-the-haystack test (Kamradt, 2023) is a popular test for the long-context retrieval abilities of language models. Besides the standard mode where the "needle" only includes the answer to be retrieved, we also use an "easy mode" (Qin et al., 2024a) where the "needle" placed in the context includes both the question and the answer. This easy mode is particularly suitable for base models that have not been instruction-tuned. Full details, including the prompts used, are in Appendix B.3.

In Figure 4, we show the results for different models. HGRN2 performs even worse than Mamba-2 and we leave it to Appendix F.5. As shown in Figure 4, FoX achieves near-perfect needle retrieval *within the training context length* in all cases. Transformer (Pro) and Transformer (LLaMA) also have perfect accuracy within the training context length in the easy mode, though they sometimes fail in the standard mode.[3] In contrast, Mamba-2 and DeltaNet (and also HGRN2 in Appendix F.5) perform poorly even within the training context length. FoX (Pro), FoX (LLaMA), and Transformer (Pro) also partially extrapolate beyond the training context length. This is expected given their per-token loss pattern beyond the training context length (see Figure 2). However, we find that the extrapolation behaviors of these models could be hyperparameter-dependent. For example, in Figure 5, we show that for FoX (Pro), the needle retrieval results and the per-token loss slope beyond the training context length vary depending on the number of training tokens and learning rates. In particular, we find that *more training tokens often leads to* worse *extrapolation*, indicating that the models may be gradually "overfitting" to their training context length during training.

### 4.4 DOWNSTREAM TASKS

We use two sets of downstream tasks: a set of short-context tasks from LM-evaluation-harness (Gao et al., 2024) and a set of long-context tasks from LongBench (Bai et al., 2023).

---

[3]Note these are small models without instruction-tuning. We expect that with more parameters/training tokens or instruction-tuning Transformers should also have perfect accuracy within the training context length.

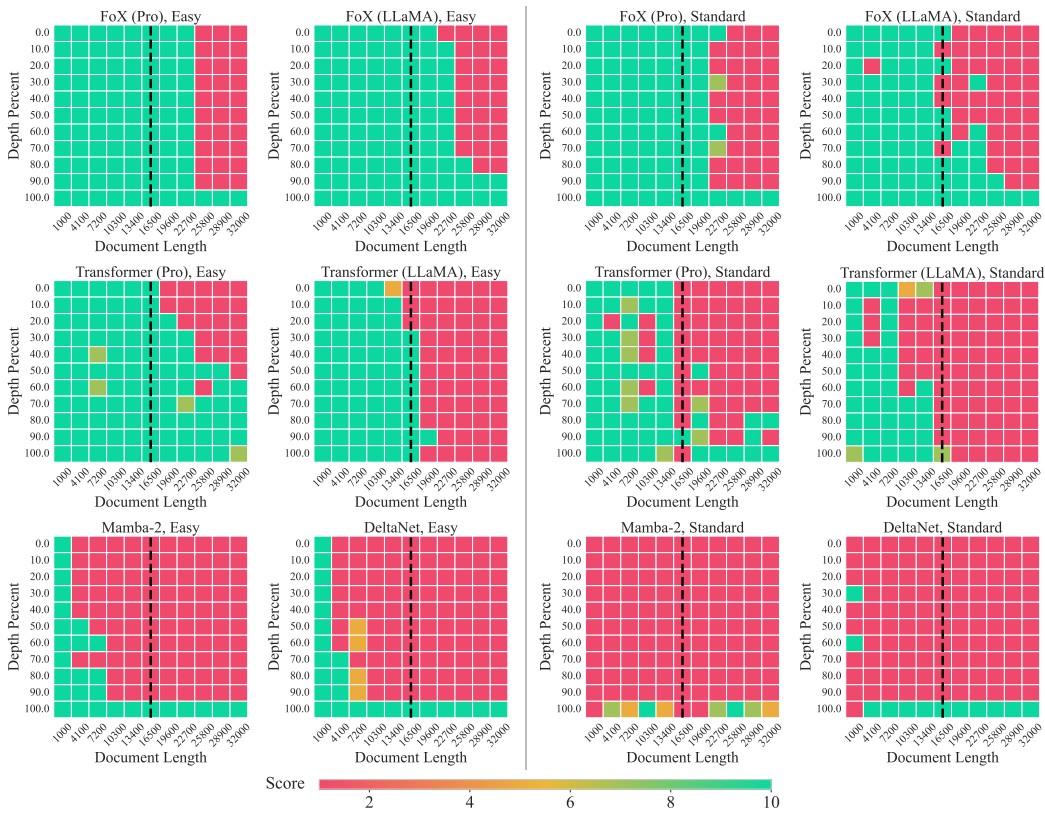

Figure 4: Needle-in-the-haystack analysis for different models. We show results for the easy mode on the left and the standard mode on the right. The results are scored on a scale of 1 to 10 by GPT-4o-2024-08-06. The vertical dashed line indicates the training context length.

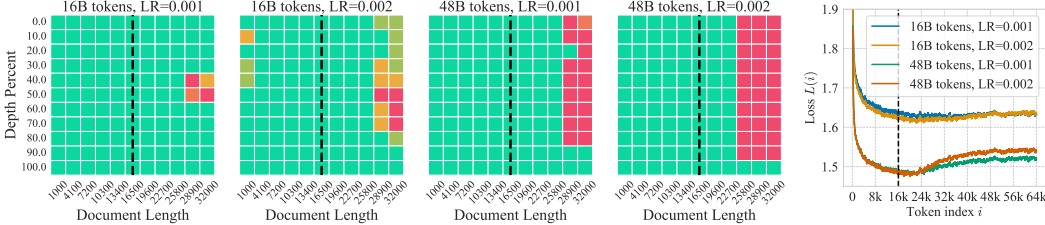

Figure 5: FoX (Pro) easy mode needle-in-the-haystack results and per-token loss for different numbers of training tokens and learning rates. The vertical dashed line indicates the training context length. More results can be found in Appendix F.7.

Table 1: Evaluation results on LM-eval-harness. All models have roughly 760M non-embedding parameters and are trained on roughly 48B tokens on LongCrawl64. "acc-n" means length-normalized accuracy. Bold and underlined numbers indicate the best and the second best results, respectively.

| Model | Wiki. ppl↓ | LMB. ppl↓ | LMB. acc↑ | PIQA acc↑ | Hella. acc-n↑ | Wino. acc↑ | ARC-e acc↑ | ARC-c acc-n↑ | COPA acc↑ | OBQA acc-n↑ | SciQA acc↑ | BoolQ acc↑ | Avg ↑ |
|---|---|---|---|---|---|---|---|---|---|---|---|---|---|
| FoX (Pro) | **23.04** | **14.91** | **42.75** | **64.09** | **38.39** | **52.33** | **52.23** | **26.54** | **71.00** | 29.80 | **85.10** | 46.57 | **50.88** |
| Transformer (Pro) | 24.12 | 16.16 | 41.47 | 64.04 | 36.60 | 49.72 | 51.98 | 25.26 | 62.00 | 29.20 | 82.80 | **60.86** | 50.39 |
| FoX (LLaMA) | 26.45 | 18.27 | 40.17 | 63.44 | 35.17 | 51.78 | 49.66 | 25.09 | 69.00 | 28.00 | 81.90 | 54.04 | 49.82 |
| Transformer (LLaMA) | 28.14 | 22.34 | 38.27 | 63.22 | 34.20 | 49.49 | 47.98 | 24.49 | 66.00 | 29.40 | 78.90 | 58.93 | 49.09 |
| Mamba-2 | 28.20 | 21.05 | 36.50 | 63.17 | 35.86 | 50.59 | 49.96 | 25.60 | **71.00** | **31.00** | 80.90 | 57.49 | 50.21 |
| HGRN2 | 30.57 | 20.14 | 38.60 | 63.49 | 34.94 | 51.78 | 50.13 | 25.51 | 66.00 | 30.00 | 75.60 | 58.41 | 49.45 |
| DeltaNet | 29.17 | 29.14 | 34.27 | 62.73 | 33.28 | 50.28 | 47.39 | 24.32 | 70.00 | 29.00 | 74.30 | 54.37 | 47.99 |

Table 2: Evaluation results on LongBench. All models have roughly 760M non-embedding parameters and are trained on roughly 48B tokens on LongCrawl64. Bold and underlined numbers indicate the best and the second-best results, respectively.

| Model | Single-Document QA | | | Multi-Document QA | | | Summarization | | | Few-shot Learning | | | Code | |
| --- | --- | --- | --- | --- | --- | --- | --- | --- | --- | --- | --- | --- | --- | --- |
| | NarrativeQA | Qasper | MFQA | HotpotQA | 2WikiMQA | Musique | GovReport | QMSum | MultiNews | TREC | TriviaQA | SamSum | LCC | RepoBench-P |
| FoX (Pro) | **13.38** | 18.88 | **28.73** | 15.27 | **25.39** | **6.49** | **22.71** | **13.51** | 12.27 | **63.5** | 37.36 | 22.74 | 10.9 | 9.1 |
| Transformer (Pro) | 11.42 | **21.54** | 22.89 | **19.58** | 22.65 | 6.09 | 21.92 | 10.7 | 8.11 | 55.0 | **40.67** | **30.66** | 10.79 | 14.25 |
| FoX (LLaMA) | 10.47 | 14.81 | 24.71 | 13.03 | 21.58 | 5.25 | 20.05 | 10.97 | 4.86 | 61.5 | 34.48 | 19.13 | 7.69 | 8.12 |
| Transformer (LLaMA) | 11.11 | 13.5 | 21.52 | 9.42 | 21.33 | 4.32 | 18.53 | 8.43 | 10.99 | 51.5 | 28.41 | 19.17 | 8.21 | 14.06 |
| Mamba-2 | 10.65 | 11.26 | 16.98 | 11.59 | 16.69 | 5.0 | 9.31 | 11.22 | 10.89 | 28.5 | 15.6 | 16.19 | 12.07 | 15.17 |
| HGRN2 | 8.78 | 10.94 | 18.66 | 7.78 | 15.29 | 4.32 | 6.13 | 12.19 | 7.83 | 16.5 | 14.46 | 6.37 | **18.17** | **16.62** |
| DeltaNet | 9.36 | 9.76 | 16.49 | 6.57 | 15.09 | 2.76 | 8.19 | 12.3 | 7.62 | 35.5 | 17.57 | 18.42 | 12.24 | 3.94 |

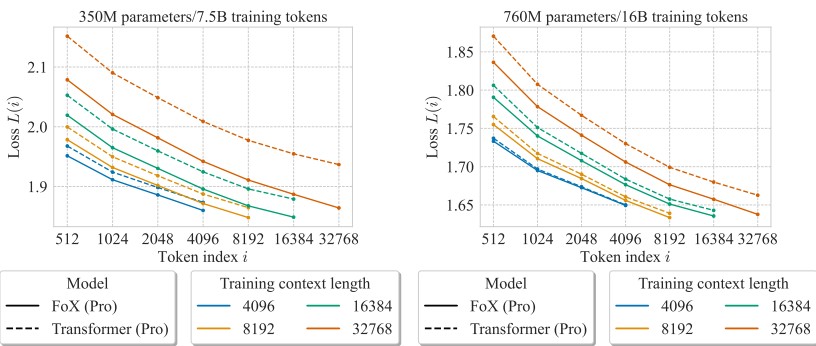

Figure 6: Per-token loss given different model sizes, numbers of training tokens, and training context lengths. At each token index $i$, we report the averaged loss over a window of 101 centered at $i$. We only show results within the training context length to reduce visual clutter. See Appendix F.6 for additional results, including length extrapolation and 125M-parameter model results.

**Short-context tasks** We use Wikitext (Merity et al., 2016), LAMBADA (Paperno et al., 2016), PiQA (Bisk et al., 2020), HellaSwag (Zellers et al., 2019), WinoGrande (Zellers et al., 2019), ARC-easy, ARC-challenge (Clark et al., 2018), Copa (Roemmele et al., 2011), SciQA (Auer et al., 2023), OpenbookQA (Mihaylov et al., 2018), and BoolQA (Clark et al., 2019). Following Yang et al. (2023), we report perplexity for Wikitext and LAMBADA, length-normalized accuracy for HellaSwag, ARC-challenge, and OpenbookQA, and accuracy for all other tasks (we also report accuracy for LAMBADA). All results are zero-shot. As shown in Table 1, FoX outperforms the Transformer with either architecture. FoX (Pro) performs the best among all models.

**Long-context tasks** We use 14 tasks from LongBench: HotpotQA (Yang et al., 2018), 2WikiMultihopQA (Ho et al., 2020), MuSiQue (Trivedi et al., 2022), MultiFieldQA-en, NarrativeQA (Kočiskỳ et al., 2018), Qasper (Dasigi et al., 2021), GovReport (Huang et al., 2021), QMSum (Zhong et al., 2021), MultiNews (Fabbri et al., 2019), TriviaQA (Joshi et al., 2017), SAMSum (Gliwa et al., 2019), TREC (Li & Roth, 2002), LCC (Guo et al., 2023), and RepoBench-P (Liu et al., 2023). We use the default metrics of LongBench for different tasks, which are either F1, Rough-L, accuracy, or edit similarity. As shown in Table 2, with either architecture, FoX performs on par with the Transformer and better than the recurrent sequence models.

## 4.5 ANALYSES

We present three sets of analyses. First, we study how the advantages of FoX over the Transformer vary with model size and training context length. Second, we investigate the contribution of each component in FoX and how RoPE affects performance. Finally, we study the importance of using a forget gate that is data-dependent. For these experiments, we use either 760M-parameter models trained on roughly 16B tokens or 360M-parameter models trained on roughly 7.5B tokens. Experimental details can be found in Appendix B.

Table 3: Ablation experiments for FoX. We use 360M-parameter models trained on 7.5B tokens on LongCrawl64. The perplexity is measured over a validation context length of 16384 tokens. For the bottom half, all addition (+) or removal (-) of components are relative to FoX (Pro).

| Model | RoPE | Forget gate | QK-norm | Output gate | Output norm | KV-shift | Perplexity |
|---|---|---|---|---|---|---|---|
| Transformer (LLaMA) w/o RoPE | | | | | | | 29.30 |
| Transformer (LLaMA) | ✓ | | | | | | 7.49 |
| | ✓ | ✓ | | | | | 7.19 |
| FoX (LLaMA) | | ✓ | | | | | 7.25 |
| | | ✓ | ✓ | | | | 7.08 |
| | | ✓ | ✓ | ✓ | | | 6.88 |
| | | ✓ | ✓ | ✓ | ✓ | | 6.80 |
| FoX (Pro) | | ✓ | ✓ | ✓ | ✓ | ✓ | 6.62 |
| - QK-norm | | ✓ | | ✓ | ✓ | ✓ | 6.79 |
| - output gate | | ✓ | ✓ | | ✓ | ✓ | 6.86 |
| - output norm | | ✓ | ✓ | ✓ | | ✓ | 6.69 |
| - KV-shift | | ✓ | ✓ | ✓ | ✓ | | 6.80 |
| + RoPE | ✓ | ✓ | ✓ | ✓ | ✓ | ✓ | 6.63 |
| - forget gate + RoPE (i.e. Transformer (Pro)) | ✓ | | ✓ | ✓ | ✓ | ✓ | 6.82 |
| - forget gate | | | ✓ | ✓ | ✓ | ✓ | 7.40 |

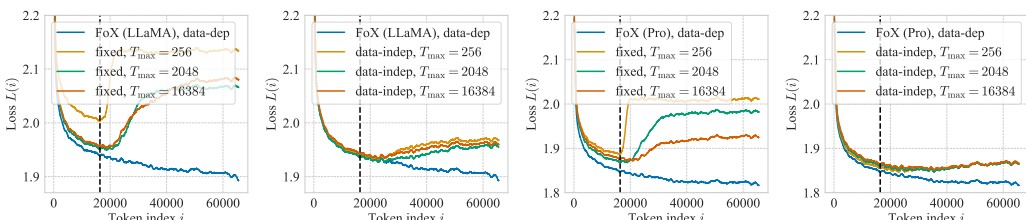

Figure 7: Data-dependent forget gate (data-dep) vs. data-independent (data-indep) and fixed forget gate. (**left** and **middle-left**) Comparison using the LLaMA architecture. (**middle-right** and **right**) Comparison using the Pro architecture. We use 360M-parameter models trained on roughly 7.5B tokens on LongCrawl64. All per-token loss curves are smoothed with a moving average sliding window of 1001 tokens. The vertical dashed line indicates the training context length.

**Model size and training context length**   In Figure 6, we show the per-token loss for two different model sizes (trained on different numbers of tokens) and several training context lengths for FoX (Pro) and Transformer (Pro). As shown in Figure 6, the advantages of FoX over Transformer (1) increase as we increase the training context length and (2) decrease as we increase the model size (and training tokens). This indicates that the advantages of having a forget gate might depend on the *ratio* between the model size and the training context length, as larger models can better model long contexts, and thus forgetting may be less important. We also note that long-context training damages short-context performance, which is a known effect (Ren et al., 2024; Sun et al., 2024) likely due to reduced document diversity within training batches.

**Component analysis**   We present both (1) an "incremental" style analysis where we incrementally add/remove components from Transformer (LLaMA) to obtain the complete FoX (Pro) model and (2) a "perturbation" style analysis where we add/remove components from FoX (Pro). The results are shown in Table 3. First, as mentioned previously, adding RoPE to FoX (LLaMA) and FoX (Pro) results in minor and no improvement, respectively. Second, both types of analyses show that all components in FoX contribute positively. Also note that models that use neither forget gates nor RoPE perform poorly (the first and the last row of the table).

**Data-independent and fixed forget gates**   To show the importance of using a forget gate that is data-dependent, we test a data-*independent* forget gate $f_t^{(h)} = \sigma(b^{(h)})$, where the superscript $(h)$ means for the $h$-th head. We also test a forget gate that has fixed values (i.e., $f_t^{(h)} = \sigma(b^{(h)})$, but we do not update $b^{(h)}$). As mentioned in Section 3, using a fixed forget gate is equivalent to ALiBi. For these data-independent forget gate designs, we find it crucial to initialize $b^{(h)}$ properly. In particular, we intialize $\{b^{(h)}\}_{h=1}^{H}$ for the $H$ heads with two hyperparameter $T_{\min}$ and $T_{\max}$ such

that using fixed forget gates with $(T_{\min}, T_{\max})$ is equivalent to ALiBi with a minimum slope $\frac{1}{T_{\max}}$ and a maximum slope $\frac{1}{T_{\min}}$. This initialization method is detailed in Appendix D.

In Figure 7, we show the per-token loss of different forget gate designs with the LLaMA and the Pro architecture. For the data-independent and the fixed forget gate designs, we set $T_{\min} = 2$ and test different values of $T_{\max}$. As shown in Figure 7, a data-dependent forget gate always has the best performance.

## 5 RELATED WORK

**Recurrent sequence models** There has been a growing interest in reviving recurrent sequence models (Katharopoulos et al., 2020; Peng et al., 2021; Gu et al., 2021; Orvieto et al., 2023; Yang et al., 2023; Gu & Dao, 2023; Katsch, 2023; De et al., 2024; Sun et al., 2024; Peng et al., 2024; Qin et al., 2024a; Dao & Gu, 2024; Beck et al., 2024; Zhang et al., 2024; Buckman et al., 2024). Many recent recurrent sequence models feature some form of the *forget gate*, which has been shown to be essential in these architectures (Qin et al., 2024b; Gu & Dao, 2023; Yang et al., 2023). Notably, GLA (Yang et al., 2023) and Mamba-2 (Dao & Gu, 2024) show that gated variants of linear attention could be written in a form similar to softmax attention, which directly inspired our work. Several works (Ma et al., 2022; 2024; Ren et al., 2024) combine recurrent layers with quadratic attention. However, unlike our method – which embeds the forget gate into the attention mechanism – in these hybrid architectures, the recurrent layers and the quadratic attention layers are independent.

**Related improvements and alternatives to softmax attention** Several positional embedding methods (Press et al., 2021; Raffel et al., 2020; Chi et al., 2022a;b) add bias terms to the attention logits based on the distances between the keys and queries, which can implement data-independent decay. LAS-attention (Zimerman & Wolf) applies multiplicative exponential decay to the attention logits. RoPE (Su et al., 2024) also has a similar decay effect that becomes stronger with increasing relative query/key distances. However, all these methods can only achieve data-*independent* decay based on the relative distances between the queries and keys. CoPE (Olsson et al., 2022) and Selective Attention (Leviathan et al., 2024) modify the current timestep's attention logit based on the sum of transformed logits from some previous timesteps. Our method differs from these in various aspects. For example, in our approach, there is no need to compute sums of transformed logits, which may come with several issues such as potential incompatibility with FlashAttention. Geometric attention (Csordás et al., 2021) and stick-breaking attention (Tan et al., 2024) use a stick-breaking process to compute the attention scores, which has a similar data-dependent decay effect to our method. These methods explore in a different direction from ours, as they seek to develop alternatives to softmax attention while our approach is only an improvement on softmax attention.

## 6 CONCLUSION

We propose the Forgetting Transformer (FoX), a Transformer variant with a forget gate. Our experiments show that FoX outperforms the Transformer and several recurrent sequence models on long-context language modeling and various downstream tasks. We also show that our Pro block design greatly outperforms the basic LLaMA architecture, with or without a forget gate. We therefore recommend that future work adopt FoX (Pro) and Transformer (Pro) as baselines in addition to the commonly used LLaMA architecture.

We discuss several limitations of our work and potential future work. First, due to our limited computing resources, our main experiments only use models up to 760M parameters, 48B tokens, and a training context length of $16384$ tokens. Thus, an important direction for future work is to test FoX at larger scales. Second, we only consider causal sequence modeling in this work. It would be interesting to extend Forgetting Attention to the non-causal case. Finally, we could potentially prune computation (e.g., KV-cache eviction) adaptively based on the forget gate values, which may greatly reduce training and inference costs.

ACKNOWLEDGMENTS

ZL thanks Songlin Yang, Jacob Buckman, Zhen Qin, Artem Zholus, Benjamin Thérien, Jonathan Pilault, and Mahan Fathi for their helpful discussion. AC acknowledges funding from Microsoft research. This research was enabled in part by the compute resources, software and technical help provided by Mila (`mila.quebec`) and the Digital Research Alliance of Canada (`alliance.can.ca`).

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

# Appendix

## Table of Contents

## A  DETAILED FoX (PRO) LAYER COMPUTATION

We describe the computation of a FoX (Pro) layer (depicted in Figure 1 right) in detail. Each FoX layer takes an input sequence $(\boldsymbol{x}_i)_{i=1}^L \in \mathbb{R}^d$ and produces an output sequence $(\boldsymbol{y}_i)_{i=1}^L \in \mathbb{R}^d$. We first describe the computation for the single head case with head dimension $d_{\text{head}}$. For each time step $t$, we first compute the key $\boldsymbol{k}_t \in \mathbb{R}^{d_{\text{head}}}$, query $\boldsymbol{q}_t \in \mathbb{R}^{d_{\text{head}}}$, value $\boldsymbol{v}_t \in \mathbb{R}^{d_{\text{head}}}$, forget gate $f_t \in \mathbb{R}$, and output gate $\boldsymbol{g}_t \in \mathbb{R}^{d_{\text{head}}}$ as follows:

$$\boldsymbol{q}_t = \text{RMSNorm}(\boldsymbol{W}_q \boldsymbol{x}_t) \tag{15}$$

$$\tilde{\boldsymbol{k}}_t = \boldsymbol{W}_k \boldsymbol{x}_t, \quad \alpha_t^{\text{key}} = \sigma(\boldsymbol{w}_k^\top \boldsymbol{x}_t) \in \mathbb{R} \tag{16}$$

$$\boldsymbol{k}_t = \text{RMSNorm}(\alpha_t^{\text{key}} \tilde{\boldsymbol{k}}_{t-1} + (1 - \alpha_t^{\text{key}}) \tilde{\boldsymbol{k}}_t) \tag{17}$$

$$\tilde{\boldsymbol{v}}_t = \boldsymbol{W}_v \boldsymbol{x}_t, \quad \alpha_t^{\text{value}} = \sigma(\boldsymbol{w}_v^\top \boldsymbol{x}_t) \in \mathbb{R} \tag{18}$$

$$\boldsymbol{v}_t = \alpha_t^{\text{value}} \tilde{\boldsymbol{v}}_{t-1} + (1 - \alpha_t^{\text{value}}) \tilde{\boldsymbol{v}}_t \tag{19}$$

$$f_t = \sigma(\boldsymbol{w}_f^\top \boldsymbol{x}_t + b_f) \in \mathbb{R} \tag{20}$$

$$\boldsymbol{g}_t = \sigma(\boldsymbol{W}_g \boldsymbol{x}_t). \tag{21}$$

This is followed by the computation of the attention output:

$$\boldsymbol{o}_i = \frac{\sum_{j=1}^i \exp(\boldsymbol{q}_i^\top \boldsymbol{k}_j + D_{ij}) \boldsymbol{v}_j}{\sum_{j=1}^i \exp(\boldsymbol{q}_i^\top \boldsymbol{k}_j + D_{ij})} \in \mathbb{R}^{d_{\text{head}}} \tag{22}$$

where $D_{ij} = \sum_{l=j+1}^i \log f_l$ with $D_{ii} = 0$ for any $i$. We then apply the output normalization, output gate, and the final output projection:

$$\boldsymbol{y}_i = \boldsymbol{W}_o(\text{RMSNorm}(\boldsymbol{o}_i) \odot \boldsymbol{g}_i) \in \mathbb{R}^d. \tag{23}$$

For the multi-head case, each head $h$ maintains an independent copy of the parameters and computes its output sequence $(\boldsymbol{y}_i^{(h)})_{i=1}^L$ independently. All RMSNorms are also applied independently to each head.[4] $\boldsymbol{y}_i^{(h)}$'s from different heads are then summed together to obtain the final output $\boldsymbol{y}_i$. Note that similar to the standard Transformer, even though the computation and parameters of different heads are conceptually independent, most computations can be implemented equivalently by properly splitting/concatenating the intermediate vectors/weight matrices of different heads.

## B  EXPERIMENTAL DETAILS

### B.1  MODEL AND TRAINING HYPERPARAMETERS

| Configuration | $n_{\text{layers}}$ | $d_{\text{model}}$ | $d_{\text{head}}$ | Peak learning rate |
|---|---|---|---|---|
| 760M params / 48B tokens | 24 | 1536 | 64 for FoX, 128 for Transformer | See Table 5 |
| 760M params / 16B tokens | 24 | 1536 | 64 for FoX, 128 for Transformer | $1 \times 10^{-3}$ |
| 360M params / 7.5B tokens | 24 | 1024 | 64 | $2 \times 10^{-3}$ |
| 125M params / 2.7B tokens | 12 | 768 | 64 | $2 \times 10^{-3}$ |

Table 4: Hyperparameters for different configurations. The head dimension $d_{\text{head}}$ is only applicable to FoX and the Transformer. We tune $d_{\text{head}}$ for the 760M-parameter FoX and Transformer models in $\{64, 128\}$. $n_{\text{layer}}$ counts the number of *blocks*. For example, for the Transformer each block contains an attention layer and an MLP layer.

We list the hyperparameters for different training configurations used in this work in Table 4. For FoX and Transformer, we follow the HuggingFace LLaMA initialization and initialize all linear layer weights and embedding parameters with $\mathcal{N}(0, 0.02^2)$. Other hyperparameters are either mentioned in the main text (Section 4.1) or follow the default values in the Flash Lienar Attention repository (Yang & Zhang, 2024)[5]. Note that our main experiments use the 760M-parameter/48B-token

---

[4]For the QK-norm implementation, we accidentally shared a single set of $d_{\text{head}}$ RMSNorm scaling parameters across different heads in each layer for our experiments (normally there should be one set of $d_{\text{head}}$ parameters for *each* head). We have verified that this has no observable impact on performance.

[5]Commit `1c5937eeeb8b0aa17bed5ee6dae345b353196bd4`.

| Model | Learning rate |
|---|---|
| FoX (Pro) | $2 \times 10^{-3}$ |
| Transformer (Pro) | $1 \times 10^{-3}$ |
| FoX (LLaMA) | $1 \times 10^{-3}$ |
| Transformer (LLaMA) | $5 \times 10^{-4}$ |
| Mamba-2 | $2 \times 10^{-3}$ |
| HGRN2 | $2 \times 10^{-3}$ |
| DeltaNet | $1 \times 10^{-3}$ |

Table 5: Peak learning rates for different models for the 760M-parameter/48B-token main experiments. These are tuned using a grid $\{1 \times 10^i, 2 \times 10^i, 5 \times 10^i\}$ with different values of $i$.

configuration. The other three configurations are for ablation studies and additional analyses. For the 760M-parameter/48B-token experiments, we tune the learning rate for each model. We also tune the head dimension for each Transformer and FoX model, along with learning rates. For the other three configurations used for ablation studies and additional analyses, the learning rates are tuned for Transformer (LLaMA) for the 16k training context length setting and transferred to other models and training context lengths. The head dimensions for these three settings are tuned for Transformer (LLaMA) and FoX (LLaMA) for the 16k training context length setting and transferred to Transformer (Pro) and FoX (Pro) and other training context lengths. The optimal hyperparameters are chosen based on the average training loss over the last $512 \times 2^{20}$ (or 512M) tokens. Note that each token is only visited once by the models during training, so there is no fundamental difference between the training and validation loss.

We do *not* share the parameters between the embedding layer and the output layer. Following the original LLaMA architecture, no bias terms are used in linear layers, except for forget gates.[6] Weight decay is not applied to RMSNorm parameters and bias terms in linear layers (again, only used for forget gates) or convolution layers. We use `bfloat16` mixed-precision training for all models.

### B.2 MODEL PARAMETERS, ESTIMATED FLOPs, AND THROUGHPUT

We report the number of (non-embedding) parameters, estimated FLOPs, and training throughput in Table 6. For the recurrent models, we estimate FLOPs using their recurrent form for simplicity. Note that the FlashAttention kernels for FoX (Pro), Transformer (Pro), and FoX (LLaMA) are implemented in Triton by us on top of Flag Attention (FlagOpen, 2023) without significant optimization, while Transformer (LLaMA) uses the official FlashAttention implementation (Dao, 2023)) in CUDA. Also, we did not focus on optimizing the efficiency of components such as QK-norm and KV-shift. We expect these four models to have similar throughput if they are properly optimized. Since we use a long training context length compared to model size, recurrent models have a significant advantage in theoretical FLOPs due to their linear complexity. Though an exact FLOPs-matched comparison would be interesting, it will require recalibrating the scaling law for the long-context setting and is beyond the scope of this work.

| Model | Params | Forward FLOPs/token | Formula for FLOPs/token | Throughput (tokens/sec) |
|---|---|---|---|---|
| FoX (Pro) | 759M | $2.72 \times 10^9$ | $2N + 2n_{\text{layer}}d_{\text{model}}L$ | 27k |
| Transformer (Pro) | 757M | $2.72 \times 10^9$ | $2N + 2n_{\text{layer}}d_{\text{model}}L$ | 30k |
| FoX (LLaMA) | 757M | $2.72 \times 10^9$ | $2N + 2n_{\text{layer}}d_{\text{model}}L$ | 30k |
| Transformer (LLaMA) | 756M | $2.72 \times 10^9$ | $2N + 2n_{\text{layer}}d_{\text{model}}L$ | 38k |
| Mamba-2 | 780M | $1.65 \times 10^9$ | $2N + 20n_{\text{layer}}d_{\text{model}}d_{\text{state}}$ | 44k |
| HGRN2 | 756M | $1.54 \times 10^9$ | $2N + 5n_{\text{layer}}d_{\text{model}}d_{\text{head}}$ | 46k |
| DeltaNet | 757M | $1.54 \times 10^9$ | $2N + 6n_{\text{layer}}d_{\text{model}}d_{\text{head}}$ | 48k |

Table 6: Number of parameters, estimated forward pass FLOPs per token, formulas for FLOPs estimation, and training throughput in tokens per second for different models. Throughput is measured on 4 NVIDIA L40S GPUs. $N$ is the number of parameters and $L$ is the training context length.

---

[6]In preliminary small-scale experiments, we do not find bias terms in forget gates to matter for performance in a statistically significant way. We keep it as it might be useful in some cases.

### B.3 NEEDLE IN THE HAYSTACK DETAILS

Our needle-in-the-haystack test is based on LongAlign (Bai et al., 2024), which is adapted from the original needle test repositoty (Kamradt, 2023) for HuggingFace[7] models. The prompt for the standard mode has the following structure:

```
[irrelevant context...]
The best thing to do in San Francisco is eat a sandwich and sit in
Dolores Park on a sunny day.
[irrelevant context...]

There is an important piece of information hidden inside the above
document. Now that you've read the document, I will quiz you about it.
Answer the following question: What is the best thing to do in San
Francisco? Answer: The best thing to do in San Francisco is
```

The easy mode is the same, except the needle placed within the context also includes the question:

```
[irrelevant context...]
What is the best thing to do in San Francisco? Answer: The best thing to
do in San Francisco is eat a sandwich and sit in Dolores Park on a sunny
day.
[irrelevant context...]

There is an important piece of information hidden inside the above
document. Now that you've read the document, I will quiz you about it.
Answer the following question: What is the best thing to do in San
Francisco? Answer: The best thing to do in San Francisco is
```

The results are scored by GPT-4o-2024-08-06 on a scale from 1 to 10.

## C EXPLANATION ON THE RELATIONSHIP BETWEEN PER-TOKEN-LOSS SLOPE AND CONTEXT UTILIZATION

To understand the relationship between the slope of the per-token loss and context utilization of the model, we first point out that LongCrawl64 applies the preprocessing of randomly "rolling" the sequences[8] to remove any position bias. This means that *when given contexts of the same length*, the difficulty of predicting tokens at different positions is roughly the same in expectation.[9] For example, in expectation, predicting the 100-th token in a sequence *given only the previous* 90 *tokens as the context* is roughly as difficult as predicting the 90-th token given the full previous 90-token context. Therefore, if $L(100) < L(90)$, it indicates that the first 10 tokens in the context contribute to the model's predictions for the 100-th token; and larger the difference $L(90) - L(100)$ is, the more these distant tokens contribute. On the other hand, if $L(100)$ is roughly the same $L(90)$ (i.e., the graph of $L(i)$ plateaus after $i = 100$), it means the first 10 tokens do not contribute to the model's prediction for the 100-th token, either because they are inherently not useful for this prediction or the model are unable to utilize them.

In summary, the slope of $L(i)$ at token position $i$ reflects how much tokens from roughly $i$ steps earlier contribute to the model's prediction at the current token position.

## D DATA-INDEPENDENT FORGET GATE INITIALIZATION

To understand our initialization for the fixed forget gate and the data-independent forget gate, we first define a function $T(b) = \frac{1}{-\log \sigma(b)}$. This function is defined such that $\sigma(b)^{T(b)} = 1/e$ is always true

---

[7] https://huggingface.co/

[8] Concretely, this can be implemented with `np.roll` with random shift value.

[9] Note that even without random rolling, given the same number of previous tokens as the context, the difficulty of token prediction at different positions may still remain relatively uniform.

(i.e., $T(b)$ is the timesteps needed to achieve a decay of $1/e$). We then initialize $b^{(h)} = b_{\text{init}}^{(h)}$ such that $T(b_{(\text{init})}^{(h)}) = \exp(\log T_{\min} + (\log T_{\max} - \log T_{\min})\frac{h-1}{H-1})$, where $T_{\min}$ and $T_{\max}$ are hyperparameters and $H$ is the number of heads. For example, if $h = 4$ and $(T_{\min}, T_{\max}) = (2, 128)$, then we have $(T(b_{\text{init}}^{(1)}), T(b_{\text{init}}^{(2)}), T(b_{\text{init}}^{(3)}), T(b_{\text{init}}^{(4)})) = (2, 8, 32, 128)$. As mentioned in the main text, A fixed forget gate with $(T_{\min}, T_{\max})$ is equivalent to ALiBi with a minimum slope $\frac{1}{T_{\max}}$ and a maximum slope $\frac{1}{T_{\min}}$. We also tested this initialization for the data-dependent forget gate but did not find it useful, so we simply initialize $\{b^{(h)}\}_{h=1}^{H}$ to zero for the data-dependent forget gate. For the data-independent and fixed forget gate, zero initialization performs extremely poorly.

## E  HARDWARE-AWARE IMPLEMENTATION OF FORGETTING ATTENTION

---

**Algorithm 1** Forgetting Attention Forward Pass

---

**Require:** Matrices $\boldsymbol{Q}, \boldsymbol{K}, \boldsymbol{V} \in \mathbb{R}^{N \times d}$, vector $\boldsymbol{c} \in \mathbb{R}^N$ in HBM, block sizes $B_c$, $B_r$.

1: Divide $\boldsymbol{Q}$ into $T_r = \left\lceil \frac{N}{B_r} \right\rceil$ blocks $\boldsymbol{Q}_1, \ldots, \boldsymbol{Q}_{T_r}$ of size $B_r \times d$ each, and divide $\boldsymbol{K}, \boldsymbol{V}$ in to $T_c = \left\lceil \frac{N}{B_c} \right\rceil$ blocks $\boldsymbol{K}_1, \ldots, \boldsymbol{K}_{T_c}$ and $\boldsymbol{V}_1, \ldots, \boldsymbol{V}_{T_c}$, of size $B_c \times d$ each.

2: Divide the output $\boldsymbol{O} \in \mathbb{R}^{N \times d}$ into $T_r$ blocks $\boldsymbol{O}_1, \ldots, \boldsymbol{O}_{T_r}$ of size $B_r \times d$ each, and divide the logsumexp $L$ into $T_r$ blocks $L_1, \ldots, L_{T_r}$ of size $B_r$ each.

3: Let $\boldsymbol{c}^q = \boldsymbol{c}$. Devide $\boldsymbol{c}^q$ into $T_r$ blocks $\boldsymbol{c}_1^q, \ldots, \boldsymbol{c}_{T_r}^q$

4: Let $\boldsymbol{c}^k = \boldsymbol{c}$. Devide $\boldsymbol{c}^k$ into $T_c$ blocks $\boldsymbol{c}_1^k, \ldots, \boldsymbol{c}_{T_c}^k$

5: **for** $1 \leq i \leq T_r$ **do**

6:    Load $\boldsymbol{Q}_i$, $\boldsymbol{c}_i^q$ from HBM to on-chip SRAM.

7:    On chip, initialize $\boldsymbol{O}_i^{(0)} = (0)_{B_r \times d} \in \mathbb{R}^{B_r \times d}, \ell_i^{(0)} = (0)_{B_r} \in \mathbb{R}^{B_r}, m_i^{(0)} = (-\infty)_{B_r} \in \mathbb{R}^{B_r}$.

8:    **for** $1 \leq j \leq T_c$ **do**

9:       Load $\boldsymbol{K}_j, \boldsymbol{V}_j$, $\boldsymbol{c}_j^k$ from HBM to on-chip SRAM.

10:       On chip, compute $\boldsymbol{S}_i^{(j)} = \boldsymbol{Q}_i \boldsymbol{K}_j^T \in \mathbb{R}^{B_r \times B_c}$.

11:       On chip, compute $\boldsymbol{D}_i^{(j)} = \boldsymbol{c}_i^q \mathbf{1}^\top - \mathbf{1}(\boldsymbol{c}_j^k)^\top \in \mathbb{R}^{B_r \times B_c}$.

12:       On chip, compute $\boldsymbol{S}_i^{(j)} = \boldsymbol{S}_i^{(j)} + \boldsymbol{D}_i^{(j)} \in \mathbb{R}^{B_r \times B_c}$.

13:       On chip, compute $\boldsymbol{S}_i^{(j)} = \text{mask}(\boldsymbol{S}_i^{(j)}, i, j) \in \mathbb{R}^{B_r \times B_c}$.

14:       On chip, compute $m_i^{(j)} = \max(m_i^{(j-1)}, \text{rowmax}(\boldsymbol{S}_i^{(j)})) \in \mathbb{R}^{B_r}$, $\tilde{\boldsymbol{P}}_i^{(j)} = \exp(\boldsymbol{S}_i^{(j)} - m_i^{(j)}) \in \mathbb{R}^{B_r \times B_c}$ (pointwise), $\ell_i^{(j)} = e^{m_i^{(j-1)} - m_i^{(j)}} \ell_i^{(j-1)} + \text{rowsum}(\tilde{\boldsymbol{P}}_i^{(j)}) \in \mathbb{R}^{B_r}$.

15:       On chip, compute $\boldsymbol{O}_i^{(j)} = \text{diag}(e^{m_i^{(j-1)} - m_i^{(j)}})^{-1} \boldsymbol{O}_i^{(j-1)} + \tilde{\boldsymbol{P}}_i^{(j)} \boldsymbol{V}_j$.

16:    **end for**

17:    On chip, compute $\boldsymbol{O}_i = \text{diag}(\ell_i^{(T_c)})^{-1} \boldsymbol{O}_i^{(T_c)}$.

18:    On chip, compute $L_i = m_i^{(T_c)} + \log(\ell_i^{(T_c)})$.

19:    Write $\boldsymbol{O}_i$ to HBM as the $i$-th block of $\boldsymbol{O}$.

20:    Write $L_i$ to HBM as the $i$-th block of $L$.

21: **end for**

22: Return the output $\boldsymbol{O}$ and the logsumexp $L$.

---

In Algorithm 1 and 2, we provide the algorithms for computing the forward pass and backward pass of Forgetting Attention in a hardware-aware way. The algorithm is reproduced from the FlashAttention-2 paper (Dao, 2023), with the changes needed to implement Forgetting Attention added and highlighted. In this algorithm, we assume that we pre-computed the cumulative sums $\boldsymbol{c} = \text{cumsum}(\log \boldsymbol{f})$. The mask operation properly sets some entries of its first operand to $-\infty$ to satisfy the causality requirement. Note for the backward pass for ease of presentation we combine

---

**Algorithm 2** Forgetting Attention Backward Pass

---

**Require:** Matrices $\boldsymbol{Q}, \boldsymbol{K}, \boldsymbol{V}, \boldsymbol{O}, \mathbf{d}\boldsymbol{O} \in \mathbb{R}^{N \times d}$ in HBM, vector $\boldsymbol{c}, \mathbf{d}\boldsymbol{c} \in \mathbb{R}^N$ , vector $L \in \mathbb{R}^N$ in HBM, block sizes $B_c, B_r$.

1: Divide $\boldsymbol{Q}$ into $T_r = \left\lceil \frac{N}{B_r} \right\rceil$ blocks $\boldsymbol{Q}_1, \ldots, \boldsymbol{Q}_{T_r}$ of size $B_r \times d$ each, and divide $\boldsymbol{K}, \boldsymbol{V}$ in to $T_c = \left\lceil \frac{N}{B_c} \right\rceil$ blocks $\boldsymbol{K}_1, \ldots, \boldsymbol{K}_{T_c}$ and $\boldsymbol{V}_1, \ldots, \boldsymbol{V}_{T_c}$, of size $B_c \times d$ each.

2: Divide $\boldsymbol{O}$ into $T_r$ blocks $\boldsymbol{O}_1, \ldots, \boldsymbol{O}_{T_r}$ of size $B_r \times d$ each, divide $\mathbf{d}\boldsymbol{O}$ into $T_r$ blocks $\mathbf{d}\boldsymbol{O}_1, \ldots, \mathbf{d}\boldsymbol{O}_{T_r}$ of size $B_r \times d$ each, and divide $L$ into $T_r$ blocks $L_i, \ldots, L_{T_r}$ of size $B_r$ each.

3: Initialize $\mathbf{d}\boldsymbol{Q} = (0)_{N \times d}$ in HBM and divide it into $T_r$ blocks $\mathbf{d}\boldsymbol{Q}_1, \ldots, \mathbf{d}\boldsymbol{Q}_{T_r}$ of size $B_r \times d$ each. Divide $\mathbf{d}\boldsymbol{K}, \mathbf{d}\boldsymbol{V} \in \mathbb{R}^{N \times d}$ in to $T_c$ blocks $\mathbf{d}\boldsymbol{K}_1, \ldots, \mathbf{d}\boldsymbol{K}_{T_c}$ and $\mathbf{d}\boldsymbol{V}_1, \ldots, \mathbf{d}\boldsymbol{V}_{T_c}$, of size $B_c \times d$ each.

4: Let $\boldsymbol{c}^q = \boldsymbol{c}^k = \boldsymbol{c}$. Devide $\boldsymbol{c}^q$ into $T_r$ blocks $\boldsymbol{c}_1^q, \ldots, \boldsymbol{c}_{T_r}^q$. Devide $\boldsymbol{c}^k$ into $T_c$ blocks $\boldsymbol{c}_1^k, \ldots, \boldsymbol{c}_{T_c}^k$.

5: Let $\mathbf{d}\boldsymbol{c}^q = \mathbf{d}\boldsymbol{c}^k = (0)_N$. Devide $\mathbf{d}\boldsymbol{c}^q$ into $T_r$ blocks $\mathbf{d}\boldsymbol{c}_1^q, \ldots, \mathbf{d}\boldsymbol{c}_{T_r}^q$. Devide $\mathbf{d}\boldsymbol{c}^k$ into $T_c$ blocks $\mathbf{d}\boldsymbol{c}_1^k, \ldots, \mathbf{d}\boldsymbol{c}_{T_c}^k$.

6: Compute $D = \mathrm{rowsum}(\mathbf{d}\boldsymbol{O} \circ \boldsymbol{O}) \in \mathbb{R}^d$ (pointwise multiply), write $D$ to HBM and divide it into $T_r$ blocks $D_1, \ldots, D_{T_r}$ of size $B_r$ each.

7: **for** $1 \le j \le T_c$ **do**

8:  Load $\boldsymbol{K}_j, \boldsymbol{V}_j, \boldsymbol{c}_j^k$ from HBM to on-chip SRAM.

9:  Initialize $\mathbf{d}\boldsymbol{K}_j = (0)_{B_c \times d}, \mathbf{d}\boldsymbol{V}_j = (0)_{B_c \times d}$ on SRAM.

10:  **for** $1 \le i \le T_r$ **do**

11:   Load $\boldsymbol{Q}_i, \boldsymbol{O}_i, \mathbf{d}\boldsymbol{O}_i, \mathbf{d}\boldsymbol{Q}_i, L_i, D_i, \boldsymbol{c}_i^q$ from HBM to on-chip SRAM.

12:   On chip, compute $\boldsymbol{S}_i^{(j)} = \boldsymbol{Q}_i \boldsymbol{K}_j^T \in \mathbb{R}^{B_r \times B_c}$.

13:   On chip, compute $\boldsymbol{D}_i^{(j)} = \boldsymbol{c}_i^q \mathbf{1}^\top - \mathbf{1}(\boldsymbol{c}_j^k)^\top \in \mathbb{R}^{B_r \times B_c}$ .

14:   On chip, compute $\boldsymbol{S}_i^{(j)} = \boldsymbol{S}_i^{(j)} + \boldsymbol{D}_i^{(j)} \in \mathbb{R}^{B_r \times B_c}$ .

15:   On chip, compute $\boldsymbol{S}_i^{(j)} = \mathrm{mask}(\boldsymbol{S}_i^{(j)}, i, j) \in \mathbb{R}^{B_r \times B_c}$.

16:   On chip, compute $\boldsymbol{P}_i^{(j)} = \exp(\boldsymbol{S}_{ij} - L_i) \in \mathbb{R}^{B_r \times B_c}$.

17:   On chip, compute $\mathbf{d}\boldsymbol{V}_j \leftarrow \mathbf{d}\boldsymbol{V}_j + (\boldsymbol{P}_i^{(j)})^\top \mathbf{d}\boldsymbol{O}_i \in \mathbb{R}^{B_c \times d}$.

18:   On chip, compute $\mathbf{d}\boldsymbol{P}_i^{(j)} = \mathbf{d}\boldsymbol{O}_i \boldsymbol{V}_j^\top \in \mathbb{R}^{B_r \times B_c}$.

19:   On chip, compute $\mathbf{d}\boldsymbol{S}_i^{(j)} = \boldsymbol{P}_i^{(j)} \circ (\mathbf{d}\boldsymbol{P}_i^{(j)} - D_i) \in \mathbb{R}^{B_r \times B_c}$.

20:   Load $\mathbf{d}\boldsymbol{Q}_i$ from HBM to SRAM, then on chip, update $\mathbf{d}\boldsymbol{Q}_i \leftarrow \mathbf{d}\boldsymbol{Q}_i + \mathbf{d}\boldsymbol{S}_i^{(j)} \boldsymbol{K}_j \in \mathbb{R}^{B_r \times d}$, and write back to HBM.

21:   Load $\mathbf{d}\boldsymbol{c}_i^q$ from HBM to SRAM, then on chip, update $\mathbf{d}\boldsymbol{c}_i^q \leftarrow \mathbf{d}\boldsymbol{c}_i^q + \mathbf{d}\boldsymbol{S}_i^{(j)} \mathbf{1} \in \mathbb{R}^{B_r}$, and write back to HBM.

22:   On chip, compute $\mathbf{d}\boldsymbol{K}_j \leftarrow \mathbf{d}\boldsymbol{K}_j + {\mathbf{d}\boldsymbol{S}_i^{(j)}}^\top \boldsymbol{Q}_i \in \mathbb{R}^{B_c \times d}$.

23:   On chip, compute $\mathbf{d}\boldsymbol{c}_j^k \leftarrow \mathbf{d}\boldsymbol{c}_j^k - {\mathbf{d}\boldsymbol{S}_i^{(j)}}^\top \mathbf{1} \in \mathbb{R}^{B_c}$ .

24:  **end for**

25:  Write $\mathbf{d}\boldsymbol{K}_j, \mathbf{d}\boldsymbol{V}_j, \mathbf{d}\boldsymbol{c}_j^k$ to HBM.

26: **end for**

27: Compute $\mathbf{d}\boldsymbol{c} = \mathbf{d}\boldsymbol{c}^q + \mathbf{d}\boldsymbol{c}^k$ .

28: Return $\mathbf{d}\boldsymbol{Q}, \mathbf{d}\boldsymbol{K}, \mathbf{d}\boldsymbol{V}, \mathbf{d}\boldsymbol{c}$ .

---

the computation of $\mathbf{d}\boldsymbol{K}, \mathbf{d}\boldsymbol{V}, \mathbf{d}\boldsymbol{c}^k$ and the computation of $\mathbf{d}\boldsymbol{Q}, \mathbf{d}\boldsymbol{c}^q$ in a single algorithm, but in practice these are computed in two different kernels for implementation simplicity.

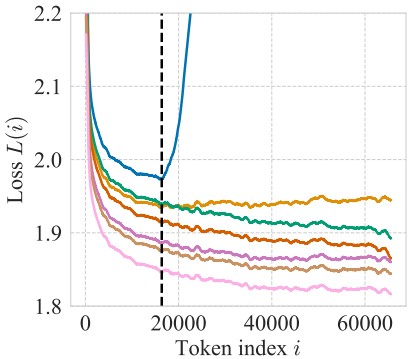

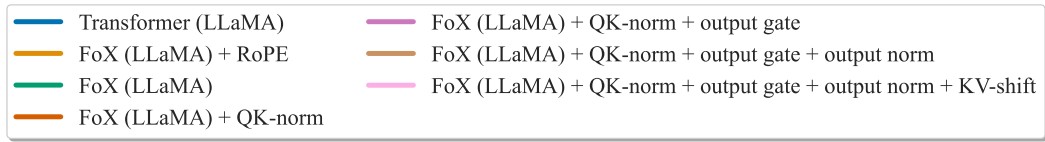

Figure 8: Per-token loss for the incremental-style ablation studies presented in Section 4.5. All models are roughly 360M parameters and are trained on roughly 7.5B tokens on LongCrawl64. The vertical line indicates the training context length.

In practice, we implement Forgetting Attention based on the Triton (OpenAI, 2021) FlashAttention implementation in Flag Attention (FlagOpen, 2023).

## F ADDITIONAL RESULTS

### F.1 PER-TOKEN LOSS FOR THE ABLATION STUDIES

In Figure 8 and Figure 9 we show the per-token loss for the ablation studies presented in Table 3 in Section 4.5. Transformer (LLaMA) without RoPE performs extremely poorly and we show it separately in Figure 10.

### F.2 TRANSFORMER (PRO) ABLATION

In Figure 11, we present a small-scale ablation study using Transformer (Pro) in the 125M-parameter/2.7B-token setting. We start with Transformer (LLaMA) and add one component at a time. Notably, we find that QK-norm seems to be helpful for length extrapolation.

### F.3 SHORT-CONTEXT TRAINING ON SLIMPAJAMA

To complement our main results in which we perform long-context training on LongCrawl64, we have also run short-context training on the more commonly used SlimPajama dataset (Soboleva et al., 2023). We follow the 340M-parameter/15B-token/2k-context-length setting in Yang et al. (2024). We also use the same hyperparameters and tokenizer as Yang et al. (2024). We train FoX and Transformer with both the LLaMA and the Pro architecture. We also test Mamba-2, the strongest recurrent sequence model in our main results.

We show the per-token loss of tested models in Figure 12 and downstream task evaluation results in Table 7. We use the same set of tasks as Yang et al. (2024) so our results can be directly compared to those of Yang et al. (2024). As shown in the results, in this short-context training setting FoX (LLaMA) does not have an advantage over the Transformer (LLaMA) except for length extrapolation, while FoX (Pro) still outperforms Transformer (Pro) in language modeling loss and downstream tasks. Note that these are small-scale experiments without extensive hyperparameter tuning (e.g., learning rate), so the results might not transfer to larger scales with proper hyperparameter tuning for each model.

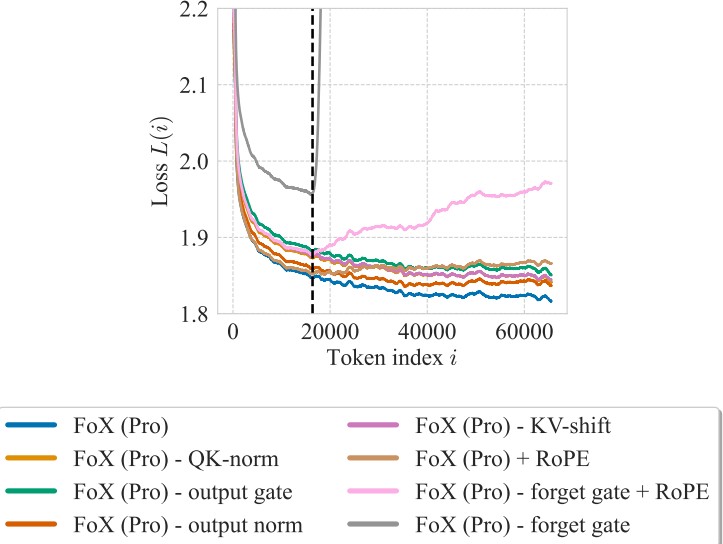

Figure 9: Per-token loss for the perturbation-style ablation studies presented in Section 4.5. All models are roughly 360M parameters and are trained on roughly 7.5B tokens on LongCrawl64. The vertical line indicates the training context length.

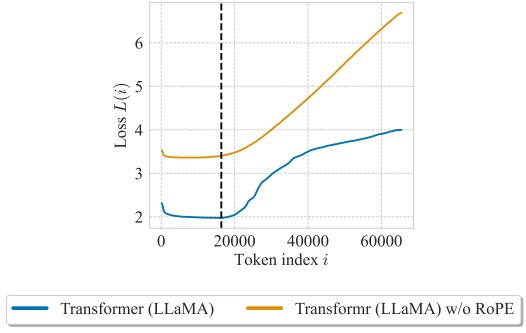

Figure 10: Removing RoPE from Transformer (LLaMA) results in poor performance. All models are roughly 360M parameters and are trained on roughly 7.5B tokens on LongCrawl64. The vertical line indicates the training context length.

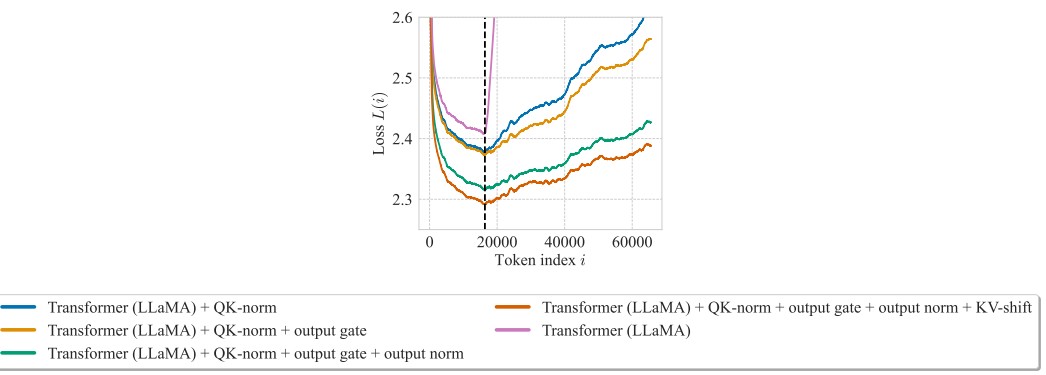

Figure 11: Incremental style ablation study for Transformer (Pro). All models are roughly 125M parameters and are trained on roughly 2.7B tokens on LongCrawl64. The vertical line indicates the training context length.

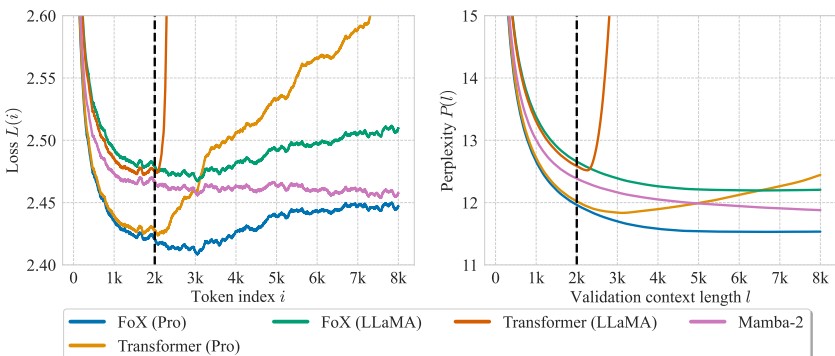

Figure 12: Results on SlimPajama with a training context length of 2048 tokens. All models have roughly 340M non-embedding parameters and are trained on roughly 15B tokens on SlimPajama. The vertical line indicates the training context length.

Table 7: Evaluation results on LM-eval-harness for models trained on SlimPajama with a training context length of 2048 tokens. All models have roughly 340M non-embedding parameters and are trained on roughly 15B tokens on SlimPajama. "acc-n" means length-normalized accuracy. Bold and underlined numbers indicate the best and the second best results, respectively. Note the results for Transformer++ and DeltaNet are from Yang et al. (2024). Note that Transformer++ from Yang et al. (2024) and Transformer (LLaMA) in our work have exactly the same architecture.

| Model | Wiki. ppl↓ | LMB. ppl↓ | LMB. acc↑ | PIQA acc↑ | Hella. acc-n↑ | Wino. acc↑ | ARC-e acc↑ | ARC-c acc-n↑ | Avg ↑ |
|---|---|---|---|---|---|---|---|---|---|
| Transformer++ (Yang et al., 2024) | 28.39 | 42.69 | 31.00 | 63.30 | 34.00 | 50.40 | 44.50 | 24.20 | 41.23 |
| DeltaNet (Yang et al., 2024) | 28.24 | 37.37 | 32.10 | 64.80 | 34.30 | **52.20** | 45.80 | 23.20 | 42.07 |
| FoX (Pro) | **25.69** | 31.98 | **35.82** | 65.61 | **36.39** | 51.07 | 45.79 | **25.09** | **43.29** |
| Transformer (Pro) | 25.92 | **31.93** | 35.01 | 65.02 | 36.09 | 50.51 | **46.42** | 23.38 | 42.74 |
| FoX (LLaMA) | 27.86 | 43.26 | 32.56 | 64.80 | 34.59 | 50.12 | 45.12 | 23.38 | 41.76 |
| Transformer (LLaMA) | 27.98 | 35.25 | 32.31 | 63.71 | 34.89 | 48.07 | 45.33 | 23.72 | 41.34 |
| Mamba-2 | 27.51 | 41.32 | 29.83 | **65.94** | 35.95 | 50.20 | 45.45 | 23.72 | 41.85 |

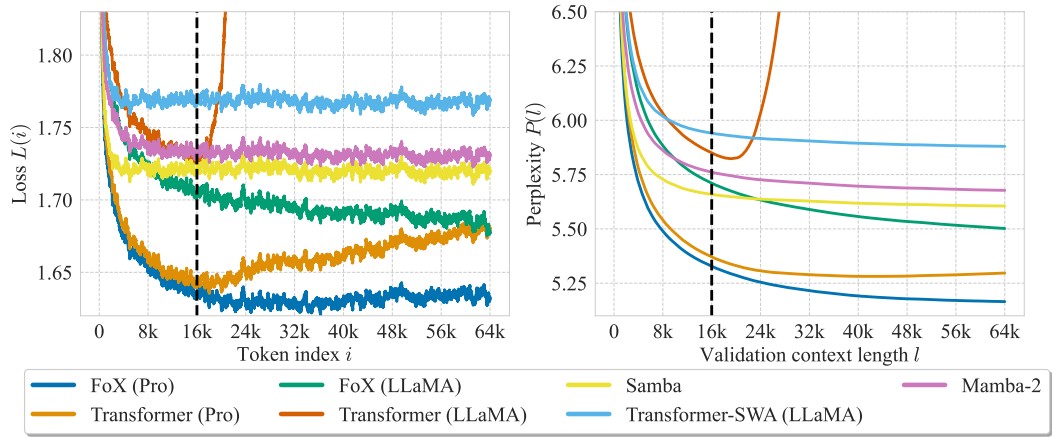

Figure 13: Additional comparison with Samba and Transformer-SWA. (**left**) Per-token loss $L(i)$ at different token position $i$. (**right**) Validation perplexity $P(l)$ over different validation context length $l$. All models have 760M parameters and are trained on roughly 16B tokens. The vertical dashed line indicates the training context length. The per-token loss is typically noisy, so we smooth the curve using a moving average sliding window of 101 tokens. In this plot $1k = 1024$.

Table 8: Evaluation results on LM-eval-harness. All models have roughly 760M non-embedding parameters and are trained on roughly 16B tokens on LongCrawl64. "acc-n" means length-normalized accuracy. Bold and underlined numbers indicate the best and the second best results, respectively.

| Model | Wiki. ppl↓ | LMB. ppl↓ | LMB. acc↑ | PIQA acc↑ | Hella. acc-n↑ | Wino. acc↑ | ARC-e acc↑ | ARC-c acc-n↑ | COPA acc↑ | OBQA acc-n↑ | SciQA acc↑ | BoolQ acc↑ | Avg ↑ |
|---|---|---|---|---|---|---|---|---|---|---|---|---|---|
| FoX (Pro) | **28.10** | **23.67** | **36.93** | **61.64** | 33.44 | 49.72 | 47.94 | 23.98 | 65.00 | 26.80 | **80.40** | 57.49 | 48.33 |
| Transformer (Pro) | 28.17 | 24.63 | 36.17 | 61.53 | **33.46** | 50.28 | 47.81 | 24.15 | **67.00** | 28.40 | 77.90 | 55.72 | 48.24 |
| FoX (LLaMA) | 31.03 | 28.41 | 34.89 | 61.21 | 32.27 | 50.51 | 46.68 | 24.06 | **67.00** | **29.60** | 77.30 | **61.07** | **48.46** |
| Transformer (LLaMA) | 32.33 | 34.41 | 32.41 | 60.94 | 31.68 | 49.96 | 45.62 | 23.63 | 64.00 | 28.60 | 74.00 | 60.06 | 47.09 |
| Samba | 31.71 | 27.78 | 34.25 | 60.45 | 32.88 | **51.70** | **49.03** | **24.32** | 61.00 | 28.20 | 78.80 | 60.58 | 48.12 |
| Transformer-SWA (LLaMA) | 33.63 | 33.04 | 33.15 | 60.01 | 31.83 | 51.14 | 46.93 | 23.38 | 62.00 | 27.40 | 76.70 | 54.62 | 46.72 |
| Mamba-2 | 33.26 | 42.38 | 27.29 | 60.83 | 32.03 | 50.67 | 46.21 | 23.55 | 64.00 | 28.40 | 76.70 | 57.61 | 46.73 |

## F.4 ADDITIONAL COMPARISON WITH SLIDING WINDOW ATTENTION AND SAMBA

In this section, we compare the standard Transformer, FoX, and Mamba-2 with a sliding-window-attention-based Transformer (Transformer-SWA). We also compare with Samba (Ren et al., 2024), a hybrid architecture combining sliding window attention and Mamba. Both Transformer-SWA and Samba use a window size of 2048. For these experiments, we use the 760M-parameter/16B-token configuration in Table 4. Note that as mentioned in Section B, all models in this configuration use the same learning rate that is tuned for Transformer (LLaMA), so the results might not be optimal for other models. We show the per-token loss, easy-mode needle-in-the-haystack experiment, short-context downstream task results, and long-context task results in Figure 13, Figure 14, Table 8, and Table 9, respectively. Though both Transformer-SWA and Samba perform well on short-context tasks, they show an early plateau in the per-token loss, which indicates that they struggle to utilize the long context. Accordingly, they perform poorly in the needle-retrieval task.

## F.5 ADDITIONAL NEEDLE-IN-THE-HAYSTACK RESULTS

In Figure 15, we show the results of the needle test for HGRN2 in the 760M-parameter/48B-token setting.

Table 9: Evalution results on LongBench. All models have roughly 760M non-embedding parameters and are trained on roughly 16B tokens on LongCrawl64. Bold and underlined numbers indicate the best and the second best results, respectively.

| | Single-Document QA | | | Multi-Document QA | | | Summarization | | | Few-shot Learning | | | Code | |
| --- | --- | --- | --- | --- | --- | --- | --- | --- | --- | --- | --- | --- | --- | --- |
| Model | NarrativeQA | Qasper | MFQA | HotpotQA | 2WikiMQA | Musique | GovReport | QMSum | MultiNews | TREC | TriviaQA | SamSum | LCC | RepoBench-P |
| FoX (Pro) | **10.48** | 12.98 | 20.62 | 6.87 | **16.2** | **5.48** | **27.51** | 10.15 | 9.27 | **63.5** | 26.97 | 18.02 | 6.34 | 3.4 |
| Transformer (Pro) | 8.67 | 13.92 | **22.45** | **9.36** | 14.21 | 5.16 | 19.88 | 10.66 | 12.23 | 52.0 | **30.18** | **25.53** | 8.37 | 10.72 |
| FoX (LLaMA) | 9.48 | **15.55** | 17.13 | 5.26 | 15.78 | 3.78 | 21.95 | 10.59 | 8.63 | 29.0 | 19.16 | 10.07 | 6.93 | 9.89 |
| Transformer (LLaMA) | 8.44 | 10.08 | 18.77 | 6.09 | 14.47 | 3.98 | 11.83 | 11.52 | **12.94** | 23.5 | 18.46 | 16.04 | 8.27 | 13.5 |
| Samba | 6.33 | 10.89 | 15.86 | 5.1 | 11.28 | 2.79 | 9.42 | 11.39 | 10.88 | 28.5 | 16.07 | 2.8 | 11.65 | **14.26** |
| Transformer-SWA (LLaMA) | 8.46 | 8.59 | 16.65 | 6.9 | 13.84 | 4.03 | 7.47 | **12.87** | 10.0 | 12.0 | 14.92 | 5.1 | **16.16** | 14.22 |

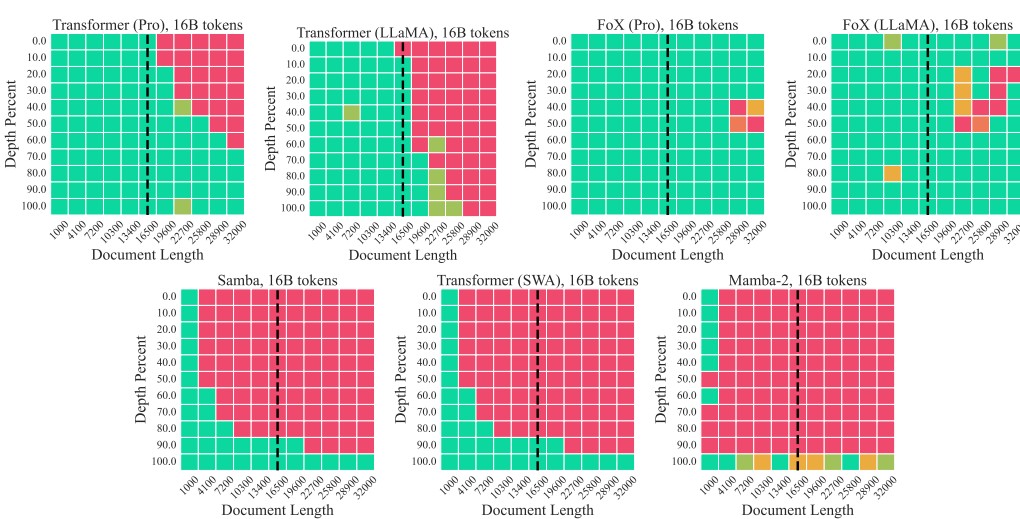

Figure 14: Easy mode needle-in-the-haystack analysis for FoX, the Transformer, Mamba-2, Samba, and the Transformer with sliding window attention. These are 760M-parameter models trained on 16B tokens on LongCrawl64. The results are scored on a scale of 1 (red) to 10 (green) by GPT-4o. The vertical dashed line indicates the training context length.

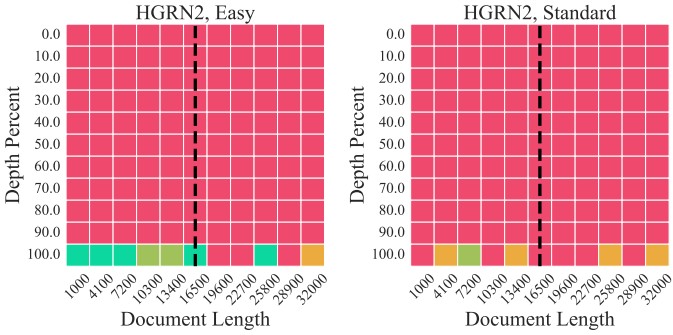

Figure 15: Needle-in-the-haystack analysis for HGRN2. The results are scored on a scale of 1 (red) to 10 (green) by GPT-4o. The vertical dashed line indicates the training context length.

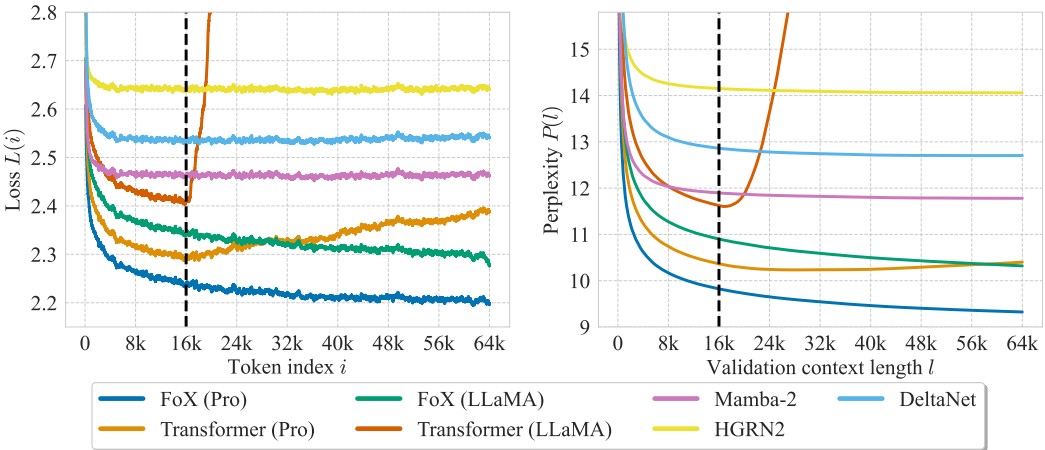

Figure 16: Results with 125M-parameter models trained on roughly 2.7B tokens. (**left**) Per-token loss $L(i)$ at different token position $i$. (**right**) Validation perplexity $P(l)$ over different validation context length $l$. The vertical dashed line indicates the training context length. The per-token loss is typically noisy, so we smooth the curve using a moving average sliding window of 101 tokens. In this plot $1k = 1024$.

### F.6 ADDITIONAL RESULTS WITH 125M-PARAM/2.7B-TOKEN, 360M-PARAM/7.5B-TOKEN, AND 760M-PARAM/16B-TOKEN TRAINING CONFIGURATIONS

Besides our main results with 760M-parameter model trained on 48B tokens, we also report per-token loss results with 125M-parameter/2.7B-token, 360M-parameter/7.5B-token, and 760M-parameter/16B-token training configurations in this section. The hyperparameters used are given in Appendix B and Table 4. Note that, as mentioned in Appendix B, the learning rates for these experiments are tuned for Transformer (LLaMA) for the 16k training context length setting and transferred to other models and training context lengths, so the reported results may not be optimal for some models (e.g., FoX typically prefers higher learning rates than the Transformer).

**Per-token loss for different models in the main experiment**  In Figure 16, Figure 17, and Figure 18, we show the per-token loss for different models given a training context length of 16k tokens for the 125M-parameter/2.7B-token, 360M-parameter/7.5B-token, and 760M-parameter/16B-token training configurations, respectively. These results are consistent with the 760M-parameter/48B-token results in Figure 2.

**Per-token loss for different training context lengths**  In Figure 19, Figure 20, and Figure 21, we show the per-token loss for the FoX and Transformer models given different training context lengths for the 125M-parameter/2.7B-token, 360M-parameter/7.5B-token, and 760M-parameter/16B-token training configurations, respectively. Consistent with the results in Figure 4.5, we see that the advantages of FoX over the Transformer (1) reduce for larger models and (2) increase for longer training context lengths.

### F.7 SENSITIVITY OF LENGTH EXTRAPOLATION BEHAVIORS TO HYPERPARAMETERS

This section presents more results on the sensitivity of length extrapolation behaviors to hyperparameters, in addition to our results in Section 4.3 and Figure 5. Figure 22 and Figure 23 show the easy-mode and standard-mode needle retrieval results for FoX (Pro) and Transformer (Pro) with different numbers of training tokens and learning rates. Figure 24 shows the corresponding per-token loss curves. As shown in these results, length extrapolation is sensitive to hyperparameters.

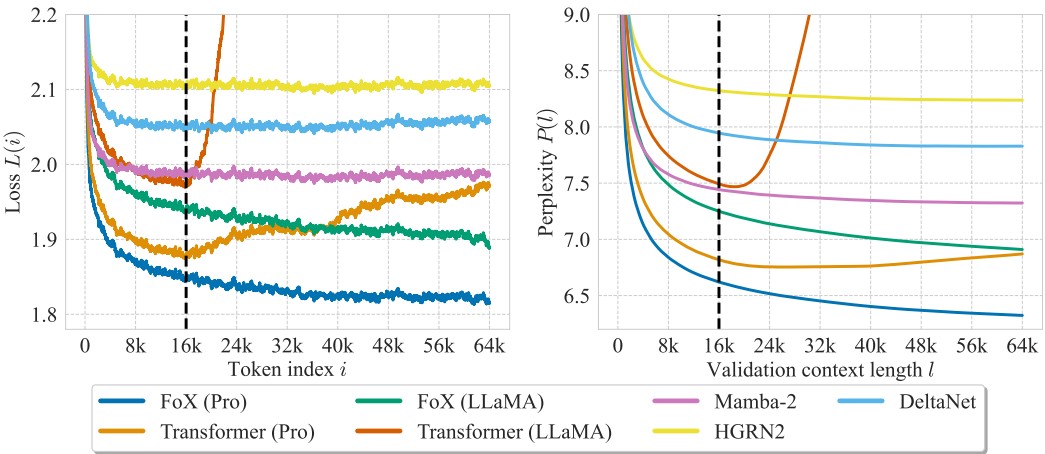

Figure 17: Results with 360M-parameter models trained on roughly 7.5B tokens. (**left**) Per-token loss $L(i)$ at different token position $i$. (**right**) Validation perplexity $P(l)$ over different validation context length $l$. The vertical dashed line indicates the training context length. The per-token loss is typically noisy, so we smooth the curve using a moving average sliding window of 101 tokens. In this plot $1k = 1024$.

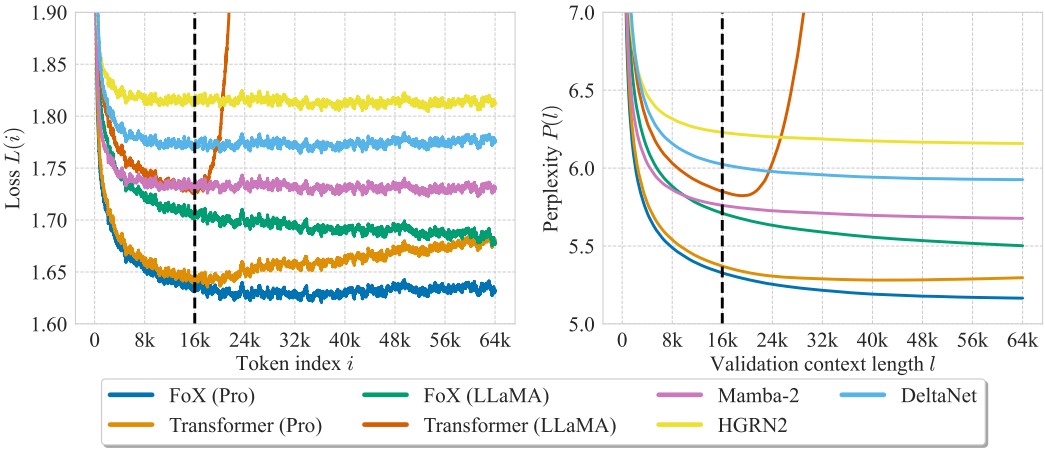

Figure 18: Results with 760M-parameter models trained on roughly 16B tokens. (**left**) Per-token loss $L(i)$ at different token position $i$. (**right**) Validation perplexity $P(l)$ over different validation context length $l$. The vertical dashed line indicates the training context length. The per-token loss is typically noisy, so we smooth the curve using a moving average sliding window of 101 tokens. In this plot $1k = 1024$.

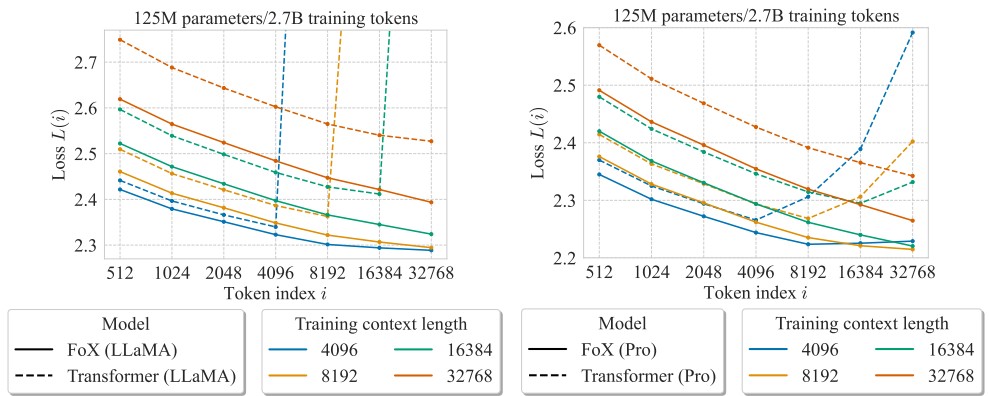

Figure 19: Per-token loss given different training context lengths for the 125M-parameter/2.7B-token setting. (**left**) Results for the LLaMA models. (**right**) Results for the Pro models. At each token index $i$, we report the averaged loss over a window of 101 centered at $i$.

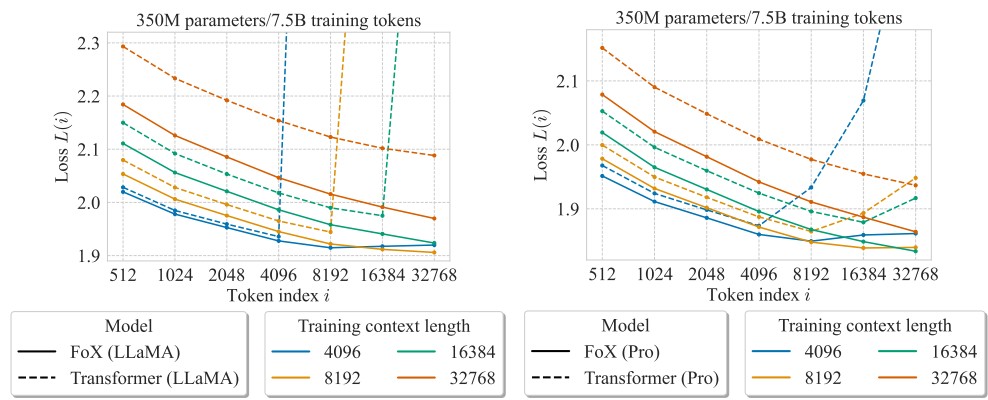

Figure 20: Per-token loss given different training context lengths for the 350M-parameter/7.5B-token setting. (**left**) Results for the LLaMA models. (**right**) Results for the Pro models. At each token index $i$, we report the averaged loss over a window of 101 centered at $i$.

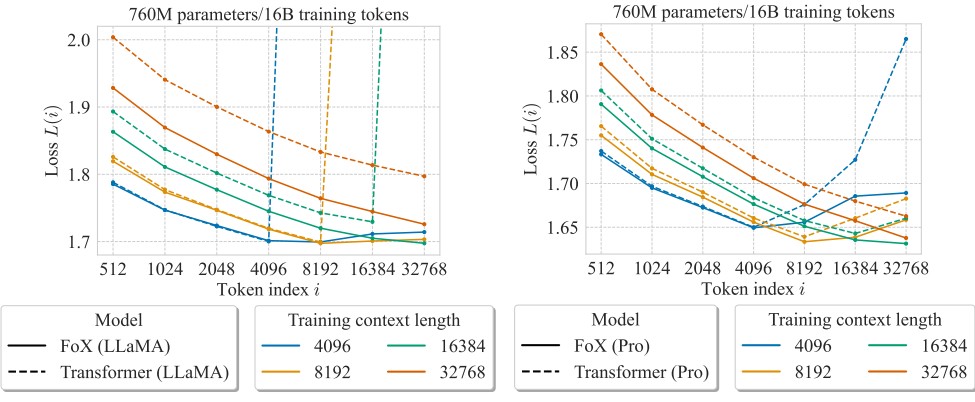

Figure 21: Per-token loss given different training context lengths for the 760M-parameter/16B token setting. (**left**) Results for the LLaMA models. (**right**) Results for the Pro models. At each token index $i$, we report the averaged loss over a window of 101 centered at $i$.

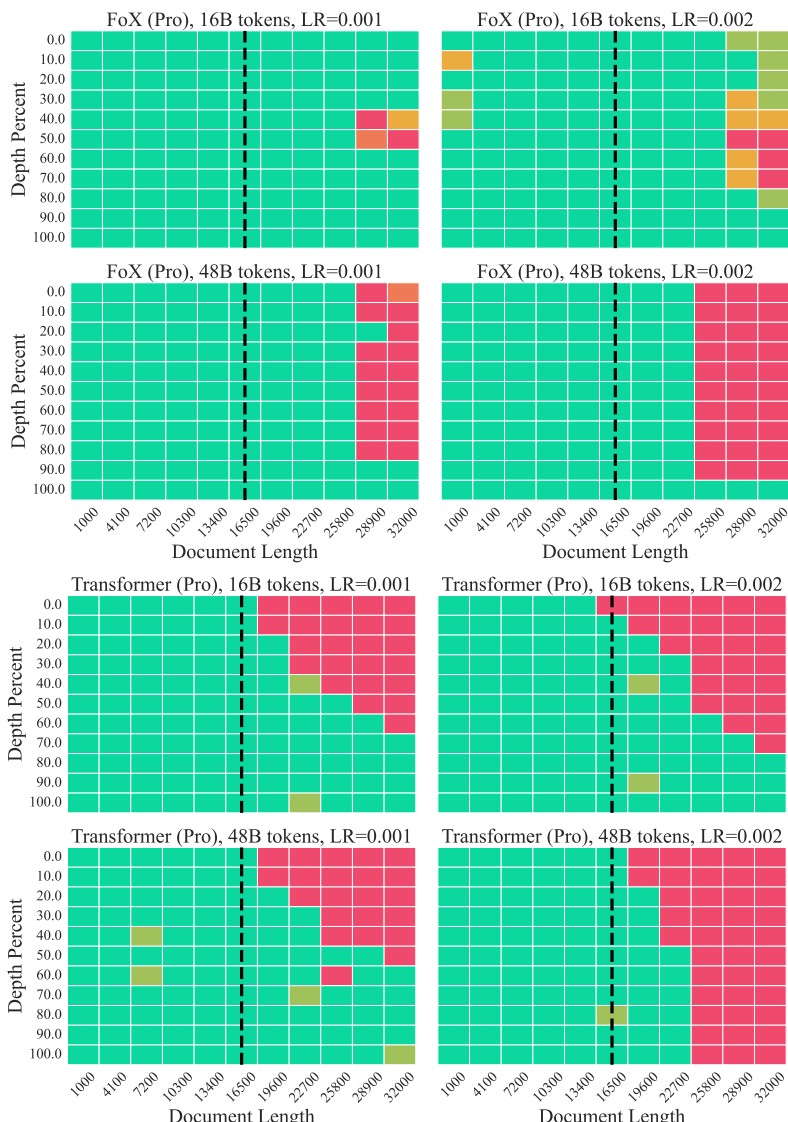

Figure 22: FoX (Pro) and Transformer (Pro) easy mode needle-in-the-haystack results for different numbers of training tokens and learning rates. The vertical dashed line indicates the training context length.

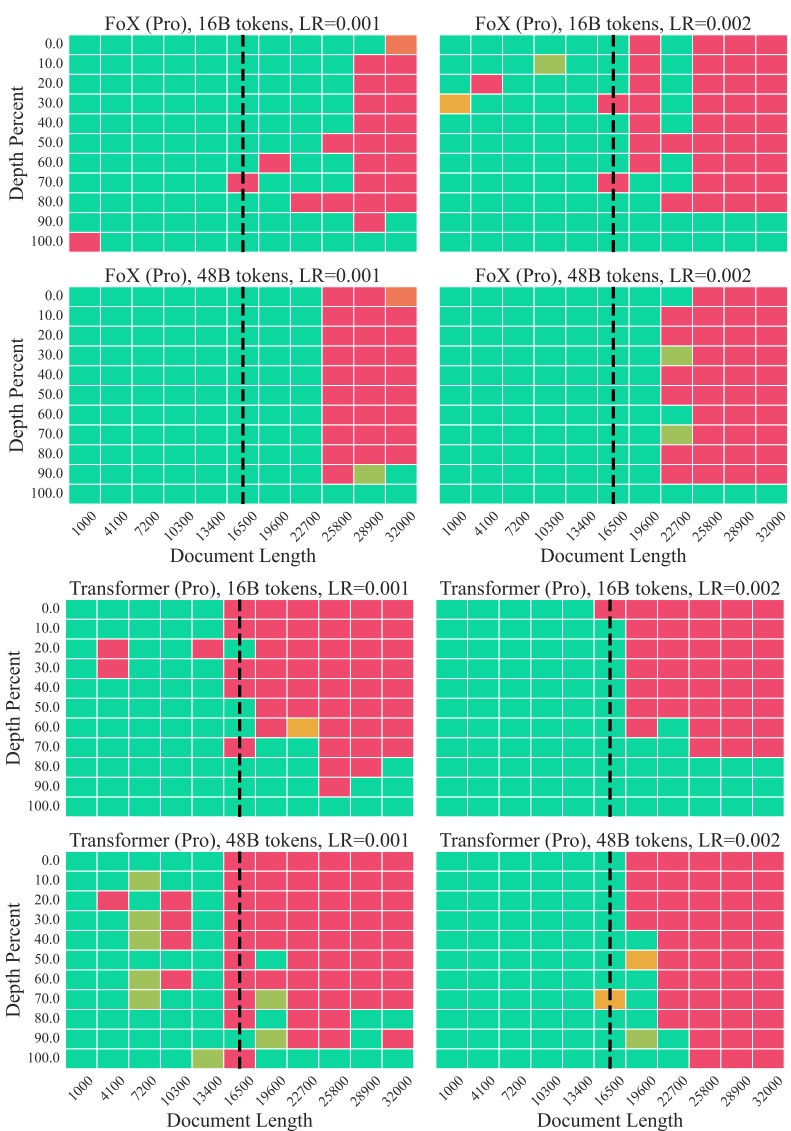

Figure 23: FoX (Pro) and Transformer (Pro) standard mode needle-in-the-haystack results for different numbers of training tokens and learning rates. The vertical dashed line indicates the training context length.

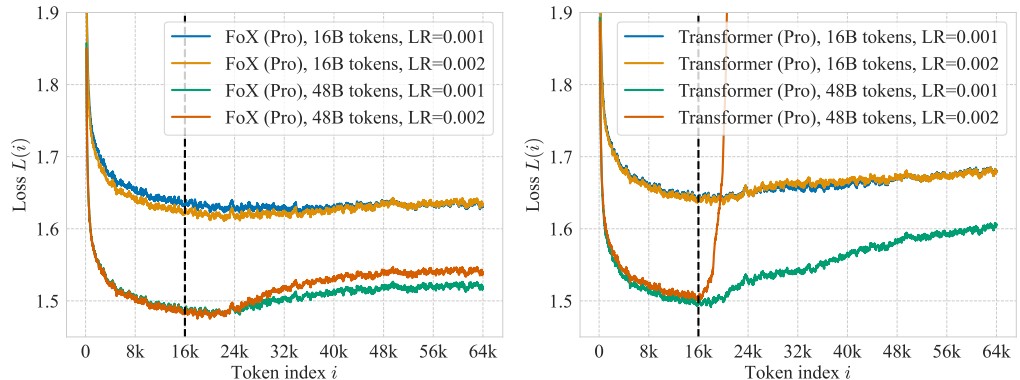

Figure 24: FoX (Pro) and Transformer (Pro) per-token loss for different numbers of training tokens and learning rates. The vertical dashed line indicates the training context length.

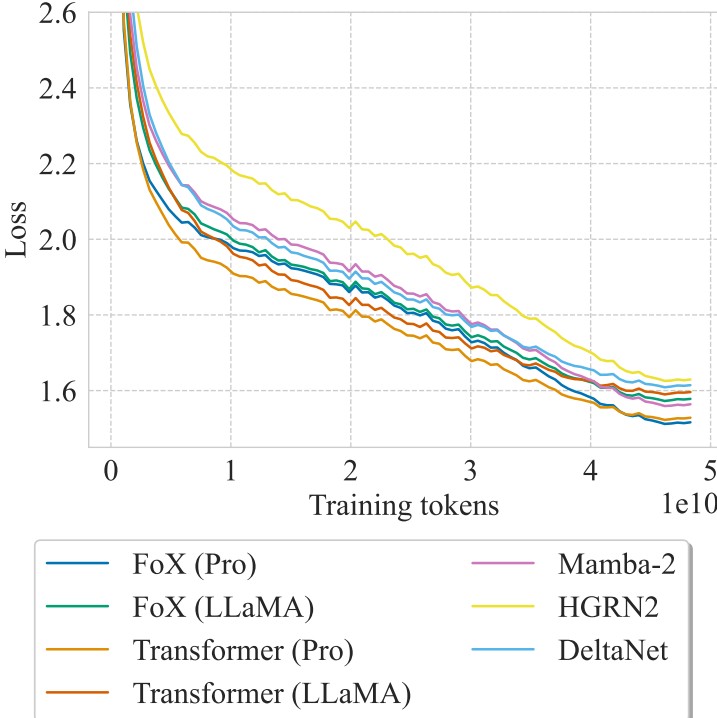

Figure 25: Training curves of different models presented in Figure 2. These curves show the training loss averaged every $512 \times 2^{20}$ tokens. All models have 760M parameters.

### F.8 TRAINING CURVES

In Figure 25, we show the training curves for all models presented in Figure 2. Note that different models use different peak learning rates, so their learning curves have different shapes.

### F.9 STABILITY ACROSS RANDOM SEEDS

Although it is computationally impractical for us to run multiple seeds for all our results, we have run three seeds for our 360M-parameter FoX (LLaMA) model to show that the variance across seeds is small. As shown in Figure 26, the variance across seeds is small.

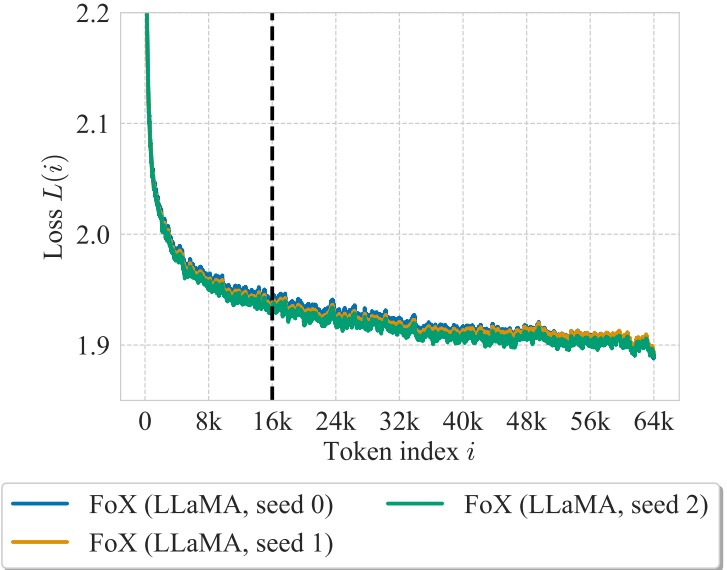

Figure 26: Result stability across seeds with 360M-parameter FoX (LLaMA). All models are trained on roughly 7.5B tokens. The vertical dashed line indicates the training context length. The per-token loss is typically noisy, so we smooth the curve using a moving average sliding window of 101 tokens. In this plot $1k = 1024$.

## F.10 ADDITIONAL VISUALIZATION OF FORGET GATE AND ATTENTION SCORE MATRICES

In Figure 27 and Figure 28, we show the forget gate matrices $\boldsymbol{F}$ and the attention score matrices $\boldsymbol{A}$ from 16 heads distributed in 4 layers. Note that since these matrices are large ($16384 \times 16384$), if only near-diagonal entries of $\boldsymbol{F}$ are non-zero the visualization will look almost all black.

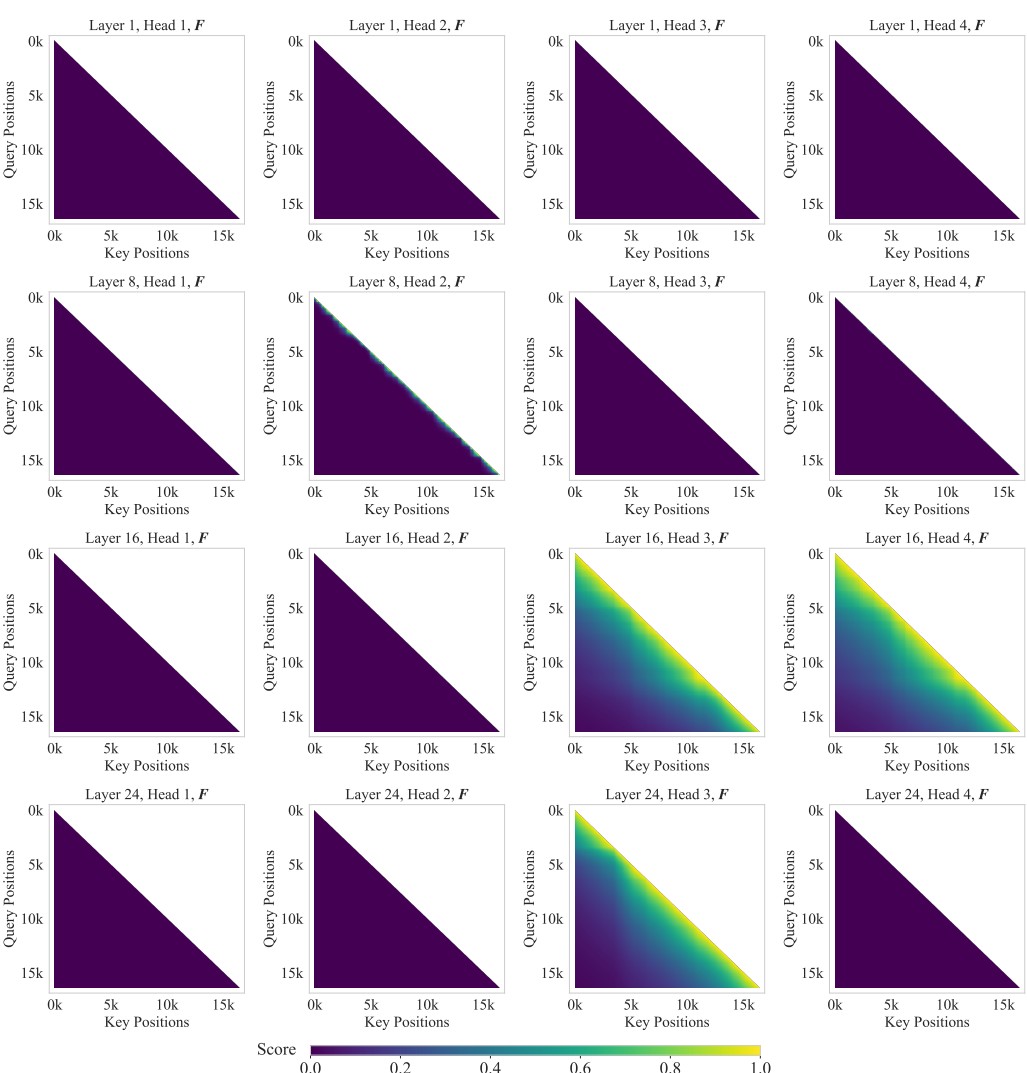

Figure 27: Visualization of the forget gate weight matrix $\boldsymbol{F}$ from 16 heads in 4 different layers. These results use FoX (Pro).

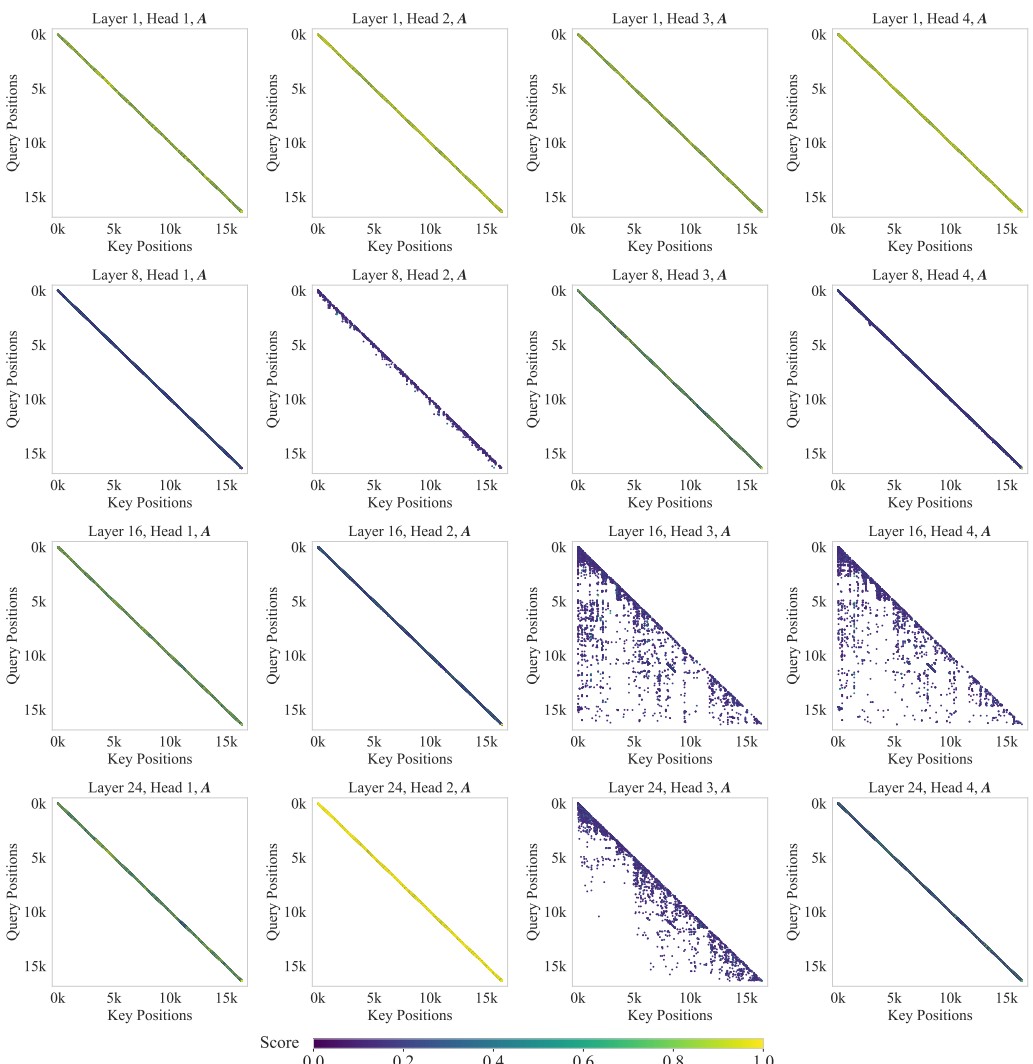

Figure 28: Visualization of the attention score matrix $A$ from 16 heads in 4 different layers. These results use FoX (Pro). Since $A$ is very sparse, we only show entries larger than 0.1.

