# OpenReview forum: "Forgetting Transformer: Softmax Attention with a Forget Gate"
_ICLR.cc/2025/Conference — ICLR 2025 Poster_

### Official Review · Reviewer_VGjT · 2024-11-01

**Soundness:** 2
**Presentation:** 3
**Contribution:** 3
**Rating:** 6
**Confidence:** 4

**Summary:**

The authors introduce the Forgetting Transformer, a transformer variant capable of dynamically focusing on various scopes of history. The mechanism is inspired by the data-dependent forget-gate in recurrent models, which are widely used in modern RNNs. The proposed method demonstrates strong length generalization and long-range capabilities, which the authors evaluated through several benchmarks, including LongCrawl64,  Needle, LongBench, and others. Finally, the authors present an efficient implementation that relies on flash attention.

**Strengths:**

1.	The proposed transformer variant is simple, elegant, and effective, demonstrating strong performance. If the authors' claims generalize beyond the specific cases presented, the potential impact could be significant.
2.	The selected benchmarks and metrics are commendable, incorporating long-range tasks across synthetic and real-world scenarios, such as Needle and LongBench, along with per-token perplexity measurements.
3.	The ablation studies in Figure 4 are essential, offering valuable insights into the method’s internal dynamics.
4.	Empirical results show notable improvements compared to the baselines.

**Weaknesses:**

I appreciate the ideas and results presented in the paper. However, I have several concerns about the informativeness, consistency, and importance of the method/results:

**W.1. Empirical Evaluation is Insufficient:**

**W.1.1. Current Insufficiency in Empirical Evaluation (Sensitivity to Hyperparameters and Robustness):** The NLP results rely on a **single** instance of the pre-trained forgetting transformer with a specific pre-training setup, which is insufficient to fully support the paper’s claims. Despite impressive performance on length generalization, context tasks, and zero-shot tasks, the robustness and **generality** of these results remain uncertain. I’m not convinced that the results are **not anecdotal**. To strengthen the evaluation, I suggest:

   (i) Include **training curves** comparing the variants presented in Figure 1.

   (ii) **Additional Datasets:** Conduct pre-training from scratch on more datasets. Even results on a smaller dataset like Wikitext-103 can be valuable. Ideally, it would benefit the community if the authors could compare their method directly with results from original papers. For example, several methods in this domain present results on Wikitext-103 (e.g., HGRN), the Pile (e.g., Pythia, Mamba, RWKV, HGRN, and others), and RedPajama.

   (iii) Report results from **multiple seeds** to ensure stability.

   (iv) **Robustness Across Models Sizes, Hyperparameters and Settings:** Show that results are not anecdotal by assessing robustness across different hyperparameters and settings, such as varying model sizes (see Mamba and RWKV, for example), other modalities, and different hyperparameters.

  W.1.2. Important details about the exact number of parameters, FLOPs, and latency are missing in the tables

___

**W.2. Missing baselines and related works:**
 Several transformer variants are similar to the proposed mechanism. The differences between these and the forgetting transformer should be discussed, and some of them should be used as baselines. Examples include:

   (i) Mega [1]: This method incorporates Exponential Moving Average before multiplying the Q and K matrices. While this mechanism is not data-dependent, it has several similarities, including manipulation of the attention matrix and adding recency bias.

   (ii) COPE [2]: This also manipulates the attention matrix and adds recency bias (in a data-dependent manner). It can be interpreted as a special forget gate.

   (iii) Selective Transformer [3]: In a very recent work, a mechanism similar to the forgetting transformer is introduced (see Section 3.3 and Equation 1).

   (iv) LAS Attention [4]: This method also adds recency bias (referred to as "local" in the paper) to the transformer by manipulating the attention matrix. It can also be interpreted as a (data-independent) forget gate.

___

I appreciate the ideas, results, and certain aspects of the empirical methodology in the paper. If my concerns are addressed during the discussion period, I will increase my score.

[1] Mega: Moving Average Equipped Gated Attention. Ma et al. ICLR23.

[2] Contextual Position Encoding: Learning to Count What's Important. Golovneva et al.

[3] Selective Attention Improves Transformer. Leviathan et al.

[4]  Viewing Transformers Through the Lens of Long Convolutions Layers. Zimerman et al. ICML24.

**Questions:**

1.	What is the impact of sharing the new parameters (w_f) across heads?
2.	It would be very interesting if standard LLMs could be improved by converting them into a forgetting transformer. For example, you could take Llama 3 7-B, convert it to the forgetting form, fine-tune it, and improve the results. This could be a breakthrough. Have you tried that? What are your thoughts on this?
3. Could you please shed more light on the inner dynamics of the forgetting transformer? For instance, it would be interesting to explore whether different heads focus on different scopes. This could be examined by measuring the F_{ij} values across heads. Additionally, a figure that visualizes the maximal forget values F_{ij} (for j=1, for example) across different positions and inputs could illustrate the forgetting trends (receptive field) learned by the model. While these aspects are partially explored in Figure 2, a more dedicated evaluation considering several inputs, layers, and heads could be very informative.
4. What are the limitations or failure cases of the forgetting transformer? Negative results could be very informative and show the differences between transformer variants.

---

> ### Author Response · Authors · 2024-11-25
> **Author Response**
>
> Thank you for your positive review! We are glad to hear that you find our method simple and elegant.
>
> First, we invite you to read our General Response to all reviewers which contains important updates, including (1) updated Transformer/Forgetting Transformer results with a better initialization and (2) a new "Pro" block design that significantly improves the performance of both the Transformer and the Forgetting Transformer.
>
>
> We address your specific concerns as follows:
>
> > (i) Include training curves comparing the variants presented in Figure 1.
>
> This is shown in Figure 18 of Appendix E.9. Note the performance gap between FoT (Pro) and Transformer (Pro) only starts to appear and increase near the end of training. We thus expect the gap to be even larger if they are trained with more tokens.
>
> > (ii) Additional Datasets
>
> We have run additional experiments on SlimPajama. We closely follow the 340M-parameter/15B-token/2k-context-length setting in the DeltaNet paper and use the same set of hyperparameters as theirs. We present our results and compare them with the official results in DeltaNet in Appendix E.3.
>
> In this setting, in terms of language modeling loss within the training context length, the Forgetting Transformer shows no advantage over the Transformer (FoT (Pro) is slightly better than Transformer (Pro) but FoT (LLaMA) is slightly worse than Transformer (LLaMA)). However, the Forgetting Transformer still shows a very clear advantage in downstream tasks (Table 5) and length extrapolation (Figure 10) over the Transformer.
>
>
> > (iii) Report results from multiple seeds to ensure stability.
>
> Although it is very challenging for us to run multiple seeds for all our results (running n seeds is n times more expensive) due to limited computational resources, we have run three seeds for our 360M-parameter Forgetting Transformer model to show that the variance across seeds is small. The results are shown in Figure 19 in Appendix E.10. Note the per-token loss curves within the training context length largely overlap. The variance beyond the training context length is larger but the behavior of different runs remains qualitatively similar. This is expected since there is no constraint on model behavior beyond the training context length during training.
>
>
> We also comment that most works that perform large-scale language modeling only run one seed, likely due to the computational demands and the fact the variance across runs is likely small at a large scale. For example, most models mentioned in our paper (Mamba-2, HGRN2, DeltaNet, Samba, RWKV, GLA) only use a single seed in their paper.
>
>
> > (iv) Robustness Across Models Sizes, Hyperparameters and Settings:
>
> We have run 125M-parameter/2.5B-token and 360M-parameter/7.5B-token experiments in Appendix E.7. The results are consistent with our main 760M-parameter/16B-token experiments. We also refer the reviewer to the SlimPajama experiments mentioned in point (ii) which use a completely different set of hyperparameters. As requested by another reviewer we have also run 760M-parameter/32B-token experiments in Appendix E.4 and the results are also consistent.
>
>
> > W.1.2. Important details about the exact number of parameters, FLOPs, and latency are missing in the tables
>
> We report the exact number of (non-embedding) parameters, estimated FLOPs, and throughput in Table 4 in Appendix B.2. Note that the attention kernels in FoT (Pro), Transformer (Pro), and FoT (LLaMA) are implemented by us in Triton while Transformer (LLaMA) uses the official Flash Attention kernel in CUDA. We expect these four models to have similar throughput if their kernels are all implemented in CUDA. Also, note that recurrent models have a huge advantage in theoretical FLOPs due to their linear complexity.  Though an exact FLOP-matched comparison would be interesting, it will require recalibrating the scaling law for the long-context setting and is beyond the scope of this work. Nevertheless, empirical evidence in the literature suggests it is unlikely that using larger models will qualitatively change the poor long-context capabilities of recurrent models (e.g., in the HGRN2 paper they train 3B-parameter HGRN2 and Mamba models and both still perform poorly in the needle-in-the-haystack task).

---

> ### Author Response · Authors · 2024-11-25
> **Author Response (continued)**
>
> > W.2. Missing baselines and related works (i) Mega, (ii) COPE (iii) Selective Transformer (iv) LAS-attention
>
> Thanks for your suggestion! We have included a discussion of these in the related work section. CoPE, Selective Transformer, and LAS-attention are not open-source so it is impractical for us to reproduce their results and compare with them during the rebuttal period. More importantly, **it is unclear whether they can be integrated into Flash Attention like Forgetting Transformer, which is essential for long-context training**. Mega is a relatively early model and has not been tested for large-scale language modeling, so instead we have compared our method with a more recent and likely much stronger baseline Samba [1]. Similar to Mega, Samba is a hybrid architecture combining sliding window attention and Mamba.
>
>
> We present the results for Samba in Appendix E.5. Though Samba performs well in short-context tasks, like other tested recurrent models it exhibits an early plateau in the per-token loss curve and performs poorly in the needle-in-the-haystack task, indicating that it struggles to use the long contexts well. It is also worse than FoT (Pro) and Transformer (Pro) in long-context tasks.
>
> > What is the impact of sharing the new parameters (w_f) across heads?
>
> We find this to perform poorly in small-scale experiments. This is not surprising because it prevents different heads from focusing on different temporal scopes.
>
> > It would be very interesting if standard LLMs could be improved by converting them into a forgetting transformer. For example, you could take Llama 3 7-B, convert it to the forgetting form, fine-tune it, and improve the results. This could be a breakthrough. Have you tried that? What are your thoughts on this?
>
> This is a great suggestion! We have not tried this but we expect this to perform well. In particular, the forget gates are data-dependent so they should learn quickly during finetuning. Also since the model can in principle learn to fall back to the standard Transformer if necessary by setting all forget gates to 1, we expect the performance to be at least as good as the unmodified model.
>
> > Could you please shed more light on the inner dynamics of the forgetting transformer? For instance, it would be interesting to explore whether different heads focus on different scopes.
>
> In Figure 21 in Appendix E.11 we show the $F$ matrices from more heads and layers. We see that different heads have different scopes. Interestingly, most heads have very local scopes (note they look almost black in visualization because only close-to-diagonal entries are non-zero and the matrices are huge). This could be very useful for efficient KV-cache pruning, where we evict previous KV-entries corresponding to low $F$ values.
>
> > What are the limitations or failure cases of the forgetting transformer? Negative results could be very informative and show the differences between transformer variants.
>
> We do not see clear failure cases of the Forgetting Transformer in our experiments. This is not surprising because, in principle, when necessary the Forgetting Transformer can learn to not forget at all and thus fall back to the standard Transformer (assuming RoPE is used). However, one failure case we can imagine is when the data contains mainly long-term dependency and very sparse or no short-term dependency. This is because **at initialization** the forget gate encourages the model to focus on short-term dependency and thus the model may not be able to pick up sufficient learning signal to gradually learn long-term dependency. This however may be fixed by special forget gate initialization that encourages long-term dependency at initialization, such as the "long-init" that we introduced in Section 4.5 (we do not find it to be useful for language modeling, though).
>
> We hope our response addresses your concerns. Feel free to let us know if you have more questions!
>
>
> [1] Ren, Liliang, et al. "Samba: Simple Hybrid State Space Models for Efficient Unlimited Context Language Modeling." arXiv preprint arXiv:2406.07522 (2024).

---

> ### Author Response · Authors · 2024-12-02
> **Author Response Reminder**
>
> Dear Reviewer,
>
> As today marks the final day for reviewers to communicate with authors, we wanted to kindly check if you have any remaining questions or feedback for us. We truly appreciate your time and effort in reviewing our work.
>
> Thank you once again for your valuable feedback!

---

> > ### Comment · Reviewer_VGjT · 2024-12-02
> >
> > I thank the authors for their detailed responses and for conducting additional experiments, which have addressed several of my concerns. I also appreciate the new empirical analyses added in response to feedback from other reviewers.
> >
> > I consider the revised manuscript to be improved, particularly in terms of methodology, robustness, clarity, and depth of insights.
> > I will carefully re-evaluate the paper after the rebuttal period to thoroughly assess its contributions.

---

### Official Review · Reviewer_vgjA · 2024-11-04

**Soundness:** 3
**Presentation:** 3
**Contribution:** 3
**Rating:** 8
**Confidence:** 4

**Summary:**

This paper proposes a modified causal attention mechanism for transformers which incorporates a _forget gate_, inspired by forget gates in recurrent neural networks. The forget gate can take the place of traditional positional encodings, so the overall architecture remains quite simple. The authors show that the forget gate significantly improves length generalization when compared to a standard transformer architecture with rotary positional encodings, and it is competitive with non-transformer architectures such as Mamba 2.

**Strengths:**

The paper presents a simple, easily implementable idea that seems to improve upon prior state of the art in autoregressive language modeling. The results on length generalization are particularly compelling when compared to RoPE.

**Weaknesses:**

The authors use QK-norm in combination with their proposed method for many experiments, finding that it significantly improves performance. The paper would be improved if there were some investigation of why QK-norm seems to be important for optimal results, especially since QK-norm was introduced in the context of Vision Transformers rather than language models. It would also be nice to see results with and without QK-norm on the standard transformer baseline, so we can see if it is actually more important to use QK-norm with Forgetting Transformers than it is with normal transformers.

**Questions:**

The exposition was clear and I have no questions at this time.

---

> ### Author Response · Authors · 2024-11-25
> **Author Response**
>
> Thank you for your positive review!
>
> First, we invite you to read our General Response to all reviewers which contains important updates, including (1) updated Transformer/Forgetting Transformer results with a better initialization and (2) a new "Pro" block design that significantly improves the performance of both the Transformer and the Forgetting Transformer.
>
> We answer your question regarding QK-norm as follows:
>
> > The paper would be improved if there were some investigation of why QK-norm seems to be important for optimal results, especially since QK-norm was introduced in the context of Vision Transformers rather than language models. It would also be nice to see results with and without QK-norm on the standard transformer baseline.
>
> In Figure 17 of Appendix E.8 we show the results for Transformer (LLaMA) + QK-norm. We see that QK-norm is equally useful for both models. Interestingly, it improves the length extrapolation of the standard Transformer. Note that QK-norm is incorporated into both the Transformer (Pro) and Forgetting Transformer (Pro) models presented in our updated main results.
>
> We comment that QK-norm has previously been used in language modeling [1][2] and is found to be very useful. Our guess for the effectiveness of QK-norm is the same as [1][2][3]: QK-norm keeps the queries and keys bounded and thus prevents the attention logits from growing uncontrollably, which will cause the attention weights to collapse to one-hot vectors. This might also explain why in our results QK-norm improves length extrapolation of the standard Transformer: extrapolation to unseen lengths might cause unusual activation values, which might result in large attention logits. We refer the reviewer to [1][2] for an analysis of the effect of QK-norm on attention logit growth.
>
> We hope our response addresses your questions. Feel free to let us know if you have more questions!
>
> [1] Wortsman, Mitchell, et al. "Small-scale proxies for large-scale transformer training instabilities. Sep 2023." URL http://arxiv. org/abs/2309.14322 v2.
>
> [2] Muennighoff, Niklas, et al. "OLMoE: Open Mixture-of-Experts Language Models." arXiv preprint arXiv:2409.02060 (2024).
>
> [3] Dehghani, Mostafa, et al. "Scaling vision transformers to 22 billion parameters." International Conference on Machine Learning. PMLR, 2023.

---

> ### Author Response · Authors · 2024-12-02
> **Author Response Reminder**
>
> Dear Reviewer,
>
> As today marks the final day for reviewers to communicate with authors, we wanted to kindly check if you have any remaining questions or feedback for us. We truly appreciate your time and effort in reviewing our work.
>
> Thank you once again for your valuable feedback!

---

### Official Review · Reviewer_bWGZ · 2024-11-06

**Soundness:** 3
**Presentation:** 3
**Contribution:** 2
**Rating:** 5
**Confidence:** 4

**Summary:**

This paper proposed to incorporate a forgetting gate, which is working as a decaying mechanism, into standard attention mechanism. The forgetting gate was directly applied to the unnormalized attention scores to down-weight long-term dependencies.

The authors pre-trained a 760M model on 16B tokens. Compared with Transformer with full attention and some recurrent architectures, such as Mamba2, HGRN2 and DeltaNet, the proposed Forgetting Transformer architecture obtained better performance on long-context retrieval task such as needle-in-haystack, and other widely used long- and short-context benchmarks.

**Strengths:**

The paper is well-written, and the proposed architecture is well-motivated and easy to implement.

**Weaknesses:**

However, there are weaknesses that the authors need to address:

1. The scale of training dataset is relatively small (16B tokens).

2. The forgetting gate works as a decaying mechanism to down-weight historical contextual information. Hence, there are some important baselines missed in the experimental comparisons. First, one straight-forward baseline is Transformer with sliding window attention. The second is chunk-wise attention with moving average mechanism (such as Megalodon).

**Questions:**

Why does the impact QK-norm on Forgetting Transformer vary on different tasks?

What are the initialization methods for the parameter $w_f$ and $b_f$?

---

> ### Author Response · Authors · 2024-11-25
> **Author Response**
>
> Thank you for your review!
>
> First, we invite you to read our General Response to all reviewers which contains important updates, including (1) updated Transformer/Forgetting Transformer results with a better initialization and (2) a new "Pro" block design that significantly improves the performance of both the Transformer and the Forgetting Transformer.
>
> We address your specific concerns as follows:
>
>
> > The scale of training dataset is relatively small (16B tokens).
>
> We recognize this scale is relatively small by the standards of modern large language models. Although it is very challenging for us to run larger-scale experiments for all our results due to limited computational resources, we have run 32B-token experiments for the Forgetting Transformer and the Transformer with the LLaMA architecture in Appendix E.4. The results are consistent with the 16B-token experiments.
>
>
> > there are some important baselines missed in the experimental comparisons. First, one straight-forward baseline is Transformer with sliding window attention. The second is chunk-wise attention with moving average mechanism (such as Megalodon).
>
> Thanks for the suggestion! For the hybrid architecture, we choose to compare with the more recent Samba model [1] instead of Megalodon since we find it very difficult to integrate the open source code of Megalodon into our codebase for technical reasons, while Samba is already implemented in the flash-linear-attention repo that we use for all our results. Samba combines sliding-window attention (SWA) with Mamba. Note that SWA is strictly better than chunk-wise attention and Mamba is strictly more expressive than moving average so Samba is likely a stronger baseline.
>
> We present the results for sliding window attention and Samba in Appendix E.5. Though both models show decent performance in short-context tasks, they show an early plateau in the per-token loss curves and perform poorly in the needle-in-the-haystack task, indicating that they struggle to use the long contexts well. They are also outperformed by FoT (Pro) and Transformer (Pro) in long-context tasks.
>
> We also comment the forget gate mechanism we propose are perfectly compatible with sliding window attention and also any hybrid architecture. Though we have not tried this, we expect it to improve these models as well.
>
>
>
> > Why does the impact QK-norm on Forgetting Transformer vary on different tasks?
>
> We are unsure of the exact reason. We believe the main function of QK-norm is keeping the queries/keys bounded, thus keeping the attention logits bounded. It is possible that this might not always be helpful in different tasks.
>
>
> > What are the initialization methods for the parameter $w_f$ and $b_f$?
>
>  $w_f$ is initialized like any other linear layer (a normal distribution with a standard deviation of $0.02$.). $b_f$ is initialized to zero. We have tried initializing $b_f$ according to the "long-init" introduced in our ablation studies but did not find it useful.
>
> We hope our response addresses your concerns. Feel free to let us know if you have more questions!
>
>
>
> [1] Ren, Liliang, et al. "Samba: Simple Hybrid State Space Models for Efficient Unlimited Context Language Modeling." arXiv preprint arXiv:2406.07522 (2024).

---

> ### Author Response · Authors · 2024-12-02
> **Author Response Reminder**
>
> Dear Reviewer,
>
> As today marks the final day for reviewers to communicate with authors, we wanted to kindly check if you have any remaining questions or feedback for us. We truly appreciate your time and effort in reviewing our work.
>
> Thank you once again for your valuable feedback!

---

### Official Review · Reviewer_Ldrv · 2024-11-10

**Soundness:** 3
**Presentation:** 4
**Contribution:** 3
**Rating:** 8
**Confidence:** 4

**Summary:**

The paper presents an addition of a forget gate to standard causal attention. In particular, the paper connects the addition of a simple multiplicative forget gate to the recurrent formulation of attention with the parallel formulation of attention and shows how it results in a simple addition of a data-dependent additive bias to softmax attention. Importantly the bias depends on all previous inputs and not just the current one. A thorough experimental evaluation shows that the proposed forgetting attention performs as well as other methods on short context tasks with possibly an improved performance on long context tasks where it performs the best in 9 out of 14 benchmarks. There are also ablations comparing the proposed forget gate to learnable or fixed decay rates where it is shown that the proposed method performs better than either option.

**Strengths:**

- The writing of the background leading to the formulation of the forgetting transformer is very good. It is a pleasure to read and it flows naturally.
- The idea is simple both to implement and understand.
- There is thorough experimental evaluation and ablation studies. I especially appreciate the comparison with data independent and fixed positional biases.

**Weaknesses:**

The paper doesn't have a lot of weaknesses. The main question is how convincing is the experimental evaluation which even though it is thorough in the sense that it evaluates most if not all possible questions it is
1. on smallish models (as mentioned by the authors themselves as a limitation)
2. lacking a baseline without RoPE or any other positional encoding

The first one is less significant given the computational requirements but the second may be important especially since the evaluation is focused heavily on really long sequences for which RoPE may not necessarily be the best choice as well as sequence length generalization for which any positional encoding will hurt performance.

The results on table 1, for the transformer's Wikipedia perplexity for instance, are quite low which calls slightly into question the validity of the conclusions. Could it be the long context training set?

**Questions:**

My main question is also slightly mentioned at the end of the weaknesses section, the transformer seems to be significantly underperforming all methods in practically every benchmark. Is this an artifact of the long context training set? Could it be undertrained or the hyperparameters not tuned sufficiently?

Similarly, how would a transformer without any positional encoding perform on these benchmarks and especially on the needle-in-the-haystack test?

---

> ### Author Response · Authors · 2024-11-25
> **Author Response**
>
> Thank you for your positive review!
>
> First, we invite you to read our General Response to all reviewers which contains important updates, including (1) updated Transformer/Forgetting Transformer results with a better initialization and (2) a new "Pro" block design that significantly improves the performance of both the Transformer and the Forgetting Transformer.
>
> We address your specific concerns as follows:
> > lacking a baseline without RoPE or any other positional encoding
>
> We have run baselines without any positional encoding or the forget gate as part of our updated ablation study. We find this always performs poorly compared to using at least one of RoPE or the forget gate. This result is consistent with either the LLaMA architecture or our new Pro architecture. See the first row and the last row of Table 3 (note for these two rows both "RoPE" and "forget gate" are not used) for results. See also the "Transformer (LLaMA) w/o RoPE" line in Figure 8 in Appendix E.1 and the "FoT (Pro) - forget gate" (equivalent to Transformer (Pro) w/o RoPE) line in Figure 7 in Appendix E.1 for the corresponding per-token loss curves.
>
> We are aware of works [1][2] that claim that not using positional encoding also works well. However, these works only use very short training context length (<2k). It might be significantly more difficult to implicitly represent positional information when the sequence is long (in our case, 16k tokens).
>
> > the transformer seems to be significantly underperforming all methods in practically every benchmark
>
> As mentioned in the General Response, we find that an important reason for this is that we use the default initialization in the flash linear attention repository (from git commits before the ICLR deadline) for the Transformer (and the Forgetting Transformer). After switching to the Huggingface LLaMA initialization, the performance of our Transformer baseline is significantly improved. This also mildly improves the performance of the Forgetting Transformer (see Figure 20 in Appendix E.11 for a comparison). We have rerun all Transformer/Forgetting Transformer results with the Huggingface LLaMA initialization.
>
> We also comment that it is well-known that long-context training can degrade short-context performance (e.g., Table 5 in [3]), likely due to reduced document diversity within training batches. Also, LongCrawl64 contains purely documents with more than 64k tokens so it might not be an ideal pretraining dataset for optimal short-context downstream task performance. To complement our long-context pretraining results, we have also performed short-context training (2k context length) on SlimPajama following the 340M-parameter/15Btoken setup in the DeltaNet paper and show that our Transformer (LLaMA) baseline matches their reported equivalent Transformer++ results. In this short-context training setting, the Forgetting Transformer has no clear advantage over the Transformer in terms of language modeling loss within the training context length but still performs better in downstream tasks and length extrapolation. Results on SlimPajama are presented in Appendix E.3.
>
>
> We hope our response addresses your concerns. Feel free to let us know if you have more questions!
>
>
>
> [1] Kazemnejad, Amirhossein, et al. "The impact of positional encoding on length generalization in transformers." in NeurIPS 2024.
>
> [2] Haviv, Adi, et al. "Transformer language models without positional encodings still learn positional information." arXiv preprint arXiv:2203.16634 (2022).
>
> [3] Ren, Liliang, et al. "Samba: Simple Hybrid State Space Models for Efficient Unlimited Context Language Modeling." arXiv preprint arXiv:2406.07522 (2024).

---

> ### Author Response · Authors · 2024-12-02
> **Author Response Reminder**
>
> Dear Reviewer,
>
> As today marks the final day for reviewers to communicate with authors, we wanted to kindly check if you have any remaining questions or feedback for us. We truly appreciate your time and effort in reviewing our work.
>
> Thank you once again for your valuable feedback!

---

### Author Response · Authors · 2024-11-25
**General Response**

# General Response

We thank all reviewers for their review and constructive feedback. In this general response, we summarize our main updates and the additional experiments we performed. In the updated paper we highlight major changes in blue. We have also included a Table of Contents in the Appendix for easy navigation.

## Updates to our main results
We have made two important updates to our main results:
1. **Updated results with better parameter initialization**: We find that switching from the default parameter initialization in the flash linear attention repository (from git commits before the ICLR deadline) that we use to the Huggingface LLaMA initialization significantly improves the performance of our Transformer baseline. This also mildly improves the performance of the Forgetting Transformer. See Figure 20 and Appendix E.11 for details and a comparison. **To ensure that we are comparing with the strongest Transformer baseline, we have rerun all the Transformer/Forgetting Transformer results with the Huggingface LLaMA initialization.** In general, this leads to a reduced performance advantage of the Forgetting Transformer over the standard Transformer. In particular, in long-context downstream tasks we find the Forgetting Transformer now performs on par with the Transformer, instead of better. However, its advantages in language modeling, length extrapolation, needle-retrieval experiments, and short-context downstream tasks remain. We have revised our claims accordingly.
2. **A new "Pro" block design**: In addition to QK-norm, we find that simple variants of several existing architectural components from recurrent sequence models (in particular, output norm, output gate, and data-dependent token shift) are also useful for the Transformer/Forgetting Transformer. We call the resulting architecture with these components the "Pro" architecture and include results for the Transformer and the Forgetting Transformer with this architecture (besides the standard LLaMA architecture). This leads to significantly improved performance for the Transformer and the Forgetting Transformer in almost all tasks. See the last paragraph of Section 3 for a description and illustration of this architecture and a thorough ablation study in Table 3 in Section 4.5. We also find the Pro architecture allows even the regular Transformer to extrapolate beyond the training context length (Figure 2), which is an interesting finding on its own. These findings with the Pro architecture constitute an additional contribution of this work.


## Additional experiments

* As suggested by Reviewer Ldrv, in Table 3 (the first row and the last row) we include baselines without any positional embeddings or forget gates. We find this to always perform poorly.
* As suggested by Reviewer bWGZ:
    * In Appendix E.4 we present results for the Forgetting Transformer and the Transformer trained with 32B tokens. The results are consistent with our main results with 16B tokens.
    * In Appendix E.5 we include additional comparison with sliding window attention and a hybrid architecture Samba. Like recurrent sequence models, these models show an early plateau in the per-token loss curve and have poor long-context retrieval performance.
* As suggested by Reviewer VGjT:
    * In Appendix E.3 we perform short-context training on SlimPajama, closely following the 340M-parameter/15B-token/2k-context length setting in the DeltaNet paper. In this case, the Forgetting Transformer has no clear advantage over the Transformer in terms of language modeling loss within the training context length but still performs better in downstream tasks and length extrapolation.
    * In Appendix E.7 we show results with 125M and 360M parameters models. These results are consistent with those of the 760M parameter models.
    * In Appendix E.10 we have run three seeds for our 360M-parameter Forgetting Transformer model to show that the variance across seeds is small.

## Minor updates

* We use a new abbreviation "FoT" for our model instead of the original "ForT".
* We have updated all our ablation studies to use 360M-parameter models instead of 125M-parameter models. All conclusions remain the same except that we find with this scale data-independent forget gates can perform as well as data-dependent forget gates within the training context length when initialized properly. The data-dependent forget gate still extrapolates better.
* As suggested by Reviewer VGjT, we have expanded the related work section to include Mega, CoPE, LAS-attention, and Selective Attention.

---

### Public Comment · ~Pooyan_Rahmanzadehgervi1 · 2025-03-11

This is a very interesting paper!

However, I find the proposed forget gate (F) in the paper and how they implement it very similar to the gated-softmax attention in Sec 4.2 of [1], which is NOT cited in this work. I also notice that Fig. 1-right in this paper is a reminiscence of Fig. 5 in [1]. Besides the motivation of applying this method, I am curious to know how they are different!


[1] Bondarenko, Yelysei, Markus Nagel, and Tijmen Blankevoort. "Quantizable transformers: Removing outliers by helping attention heads do nothing." Advances in Neural Information Processing Systems 36 (2023): 75067-75096.

---

> ### Public Comment · ~Zhixuan_Lin1 · 2025-03-11
>
> Hi Pooyan,
>
> Thank you for your question! G(x) in [1] should be seen as a per-head output gate (by the way, our Pro architecture also has an output gate, though ours is per-dimension). G(x) in [1] is applied independently to each token position. Therefore, it does not affect the time-mixing aspect of the model. This is completely different from a forget gate, which introduces temporal decay. For example, an output gate cannot make each query attend only to a local context, but a forget gate can.
>
> As for Figure 1 in our paper and Figure 5 in [1], note that any model with QKV attention (linear or not) and potentially an output gate/forget gate (e.g., most linear attention models with a forget gate/output gate) would have the structure (e.g., Figure 1 in [Gated DeltaNet](https://openreview.net/forum?id=r8H7xhYPwz)). What matters is the computation of the core time-mixing operation (in our case, the "Forgetting Attention" box in Figure 1).

---

> > ### Public Comment · ~Pooyan_Rahmanzadehgervi1 · 2025-03-11
> >
> > I agree!
> >
> > Thank you so much, Zhixuan!

---

### Meta-Review · Area_Chair_d1gv · 2024-12-19

**Metareview:**

This paper proposes adding a forget gate for attention in transformers. The proposed system amounts to a learnable data-dependant bias in the softmax. The proposed approach works well on short-context tasks and works reasonably well on long-context ones. The reviewers enjoyed the presentation and motivation, and judge the empirical validation to be sufficient. The experiments could be further improved, as written in the reviews. Some of the concerns have been lifted after the author's response.
Overall, I would like to recommend this paper for acceptance.

**Additional Comments On Reviewer Discussion:**

There was not much discussion, the reviews were almost anonymously positive. One reviewer answered after the author's rebuttal.

---

### Decision · Program_Chairs · 2025-01-22

Accept (Poster)